**Role of atmospheric aerosols in severe winter fog over Indo Gangetic Plains of India: a**
**case study**
Chandrakala Bharali[1], Mary Barth[2], Rajesh Kumar[2], Sachin D. Ghude[3], Vinayak Sinha[4], Baerbel Sinha[4]
[1] Dibrugarh University, Dibrugarh, Assam, India
[2] NSF National Centre for Atmospheric Research, Boulder, CO, US
[3] Indian Institute of Tropical Meteorology, Ministry of Earth Sciences, Pune, India
[4] Department of Earth and Environmental Sciences, Indian Institute of Science Education and Research, Mohali,
Punjab, India
**Correspondence:** chandrakalabharali@gmail.com, barthm@ucar.edu
**Abstract**
Winter fog and severe aerosol loading in the boundary layer over North India, particularly in the Indo-
Gangetic Plain (IGP), disrupt daily life of millions of people in the region. To understand better the
role of aerosol-radiation feedback on the occurrence, spatial extent, and persistence of winter fog; and
the associated aqueous chemistry in fog in the IGP, several model simulations have been performed
using the Weather Research and Forecasting model coupled with chemistry (WRF-Chem). While
WRF-Chem was able to represent the fog formation for the December 23-24, 2017 fog event over the
central IGP in comparison to station and satellite observations, the model underestimated $PM_{2.5}$
concentrations compared to the Central Pollution Control Board of India monitoring network. While
evaluating aerosol composition for fog events in IGP, we found that the WRF-Chem aerosol
composition was quite different from measurements obtained during the Winter Fog Experiment in
Delhi, with secondary aerosols, particularly chloride aerosol fraction being strongly underpredicted
(~66.6%). Missing emission sources (e.g., industry and residential burning of cow dung and trash) and
aerosol and chemistry processes need to be investigated to improve model-observation agreement. By
investigating a fog event on December 23-24, 2017 over central IGP, we found that the aerosol-
radiation feedback weakens turbulence, lowers the boundary layer height, and increases $PM_{2.5}$
concentrations and RH within the boundary layer. Factors affecting the feedback include loss of
aerosols through deposition of cloud droplets and internal mixing of absorbing and scattering aerosols.
Aqueous-phase chemistry increases the $PM_{2.5}$ concentrations, which subsequently affects the aerosol-
radiation feedback by both increased mass concentrations and aerosol sizes. With aerosol-radiation
interaction and aqueous phase chemistry, fog formation began 1-2 hours earlier and caused a longer
fog duration than when these processes were not included in the WRF-Chem simulation. The increase
in RH in both the experiments is found to be important for fog formation as it promoted the growth of
aerosol size through water uptake, increasing the fog water content over IGP. The results from this

study suggest that the aerosol-radiation feedback and secondary aerosol formation play an important role in the air quality and the intensity and lifetime of fog over IGP, yet other feedbacks, such as aerosol-cloud interactions, need to be quantified.

## 1    Introduction

The Indo-Gangetic Plain (IGP; 21°35′-32°28′N latitude. and 73°50′-89°49′E longitude) in the northern part of the Indian subcontinent is one of the most densely populated and heavily polluted regions in South Asia. The rapid population and economic growth in the IGP region over the last decade have increased air pollution over this region. This is evident from the increasing trend in AOD and $NO_2$ column concentration over India reported in recent studies (Dey and Di Girolamo, 2011; Ghude et al., 2013; Krishna Moorthy et al., 2013), which has slowed and reversed only recently (Sarkar et al., 2019). The high concentration of aerosols along the IGP and their adverse effects on human health and the environment are increasing (Ghude et al., 2016). Consequently, more than 500 million people living in the IGP breathe air that exceeds the National Ambient Air Quality Standards (NAAQS), which has reduced the life expectancy of the people (Debnath et al., 2022; Lelieveld et al., 2015). Lelieveld et al., (2015) estimated a very high number of premature deaths (0.716 million per year) linked to aerosols ($PM_{2.5}$), thus making Southeast Asia one of the largest regions affected by premature mortality globally.

One of the major environmental concerns in the IGP is the urban air quality during winter, especially over the mega-cities, e.g., Delhi, located in the north-western part of IGP (Ghude et al., 2020; Jena et al., 2021; Sengupta et al., 2022). Several urban air pollution hotspots along the IGP extend from northwest to east with monthly average $PM_{2.5}$ greater than 200 µg m$^{-3}$ (NAAQS=60µg m$^{-3}$, 24 hr average) in the winter season (Bharali et al., 2019; Krishna et al., 2019).  IGP is dominated mainly by fine mode particulates, especially over central to eastern IGP, during post-monsoon and winter (Kumar et al., 2018). Biomass burning (agricultural waste burning, domestic heating, etc.) is an important contributor to the observed high $PM_{2.5}$ loading over IGP during these seasons (Kulkarni et al., 2020; Pant et al., 2015; Pawar and Sinha, 2022; Sharma et al., 2010; Yadav et al., 2020). Delhi is affected substantially by the emissions from agricultural waste burning in the north-western states of Punjab and Haryana during the post-monsoon (October-November) season (Badarinath et al., 2009; Jethva et al., 2018; Kumar et al., 2021). Studies showed that $PM_{2.5}$ increased from ~50 µg m$^{-3}$ to as high as 300 µg m$^{-3}$ (Ojha et al., 2020), and AOD reached 0.98 with the presence of absorbing aerosols (Singh et al., 2018) during the peak biomass burning in post-monsoon.

IGP experiences fog (both radiation and advection fog) every winter after the passage of the synoptic wind system called the "Western Disturbances". The majority of fog events in the IGP during December-January are radiation fog (Deshpande et al., 2023; Ghude et al., 2023), formed due to radiative cooling of the surface. The number of low visibility days due to haze/fog formation has been

increasing significantly (Ghude et al., 2017; Jenamani, 2007; Singh and Dey, 2012), impacting socio-economic activities, e.g., aviation (Kulkarni et al., 2019). The increase in the intensity and regional extent of fog over IGP is consistent with the increasing trend in aerosol concentration due to increasing anthropogenic emissions (Sarkar et al., 2006; Syed et al., 2012).

Several factors control the formation and persistence of fog in the IGP, e.g., stable boundary layer, low temperature, availability of moisture (supplied by the Western Disturbances and irrigation activities), and the aerosol number and composition (Acharja et al., 2022; Dhangar et al., 2021). It has also been suggested that the atmospheric rivers (moisture incursion from Arabian Sea) act as a source of water vapor over IGP, which fuels the intensification of fog and haze (Verma et al., 2022) during winter. The high aerosol concentration in the boundary layer influences fog formation (Gautam et al., 2007; Safai et al., 2019) over the IGP by providing the needed cloud condensation nuclei (CCN) for activation into fog droplets. In addition, aerosols induce surface cooling by reducing solar radiation at the surface while warming the lower troposphere by absorption (Ding et al., 2016; Yu et al., 2002). A reduction in surface-reaching solar radiation by ~19% has been reported during winter over Kanpur in the IGP (Dey and Tripathi, 2007). The reduced solar flux affects the boundary layer stability and depth by suppressing the thermals and thus further increasing the surface aerosol concentration via aerosol-radiation feedback, which is very strong over the IGP (Bharali et al., 2019). Kumar et al., (2020) have shown that aerosol-radiation feedback significantly improves the accuracy of $PM_{2.5}$ and temperature forecasts in Delhi. Srivastava et al., (2018) reported that the direct aerosol forcing over polluted regions is very large with values up to $-80.0 \pm 7.2$ W m$^{-2}$ over the IGP in the winter season.

Aerosol-radiation interaction determines that the aerosol distribution is critical for the evolution of fog (Bodaballa et al., 2022; Steeneveld et al., 2015), while microphysics is important for fog formation and dispersal (Boutle et al., 2018; Maalick et al., 2016). Although the relationship between the aerosol chemical composition and aerosol activation to CCN has not been fully understood yet, studies have found that the chemical composition and mixing state of aerosols affect the hygroscopicity ($\kappa$) of aerosols (Bodaballa et al., 2022; Ma et al., 2013; Moore et al., 2012; Zhang et al., 2014a). Fog processes involve a complex interplay between local meteorology, radiation, microphysics, and aerosol chemistry, making it difficult to understand the fog lifecycle (Acharja et al., 2022; Maalick et al., 2016; Zhang et al., 2014b). There is considerable heterogeneity in the spatial and temporal aerosol properties over IGP and the poor estimates of their mixing state. Therefore, prediction of fog by weather models is still challenging with biases in fog's onset and dispersal timings.

Previous studies have focussed on the impacts of meteorological conditions, topography, or anthropogenic emissions on the poor air quality and intensification of fog during winter over IGP (e.g. Hakkim et al., 2019). However, studies on the effect of feedback induced by the aerosols on the meteorological conditions and thus on aerosol concentration are very limited over this region, except for a few above-mentioned studies which discuss how the aerosol-radiation feedback favors haze and

fog during winter. Moreover, fog can provide a medium for aqueous-phase reactions. While several
earlier studies have reported an increase in secondary aerosols during fog over IGP, a sensitivity study
examining the impact of fog on aqueous phase chemistry has not yet been done over IGP.
In the present work, we aim to find the suitable chemistry/physics as well as the meteorology
initial/boundary conditions that lead to improved simulations of fog events in the Weather Research
and Forecasting model coupled with chemistry (WRF-Chem; (Fast et al., 2006; Grell et al., 2005;
Powers et al., 2017). We also explore the role of aerosol-radiation feedback on fog properties as the
high aerosol loadings in northern India can impact the heating rates, temperature inversions, and
boundary layer height. The role of aqueous chemistry on fog properties and vice-versa is also
investigated.

**2    Methodology**
Fog formed due to radiative cooling, based on the onset time of fog and low wind speeds (Deshpande
et al., 2023 and references therein), at the surface on both $23^{rd}$ and $24^{th}$ December 2017 over a
widespread region of the IGP (Fig. 1a, b). The fog region is located over an area with high $PM_{2.5}$
anthropogenic emissions (Fig. 1c). The IGP is a large region with varying meteorology and aerosol
characteristics, therefore, it is divided into three areas, northwest (NWIGP: latitude-longitude range,
27°N-32°N,75°E-79°N), central (CIGP: latitude-longitude range, 25°N-28°N,79°E-83°E), and east
(EIGP: latitude-longitude range, 24°N-27°N, 83°E-87°E) which are marked by the black rectangles in
Fig.1c. Although biomass burning and anthropogenic emissions dominate throughout the IGP during
post-monsoon and winter season, the north-westerly wind system results in the gradient distribution of
AOD over this region. The downwind regions, CIGP and EIGP are influenced by the long-range
transport from the NWIGP, resulting in high AOD with dominant fine particulates over CIGP and
EIGP, especially during post-monsoon and winter (Kedia et al., 2014; Kumar et al., 2018). Therefore,
representative stations from each region listed in section 2.2 are considered for the sensitivity
analyses.

**2.1 Modeling**
The WRF-Chem model version 4.0.3 is used for this study. Earlier studies have successfully
used WRF-Chem to predict fog (Pithani et al., 2019) and in the study of aerosol-radiation feedback on
air quality (Kumar et al., 2020; Bharali et al., 2019) and fog (Shao et al., 2023). The model domain is
centered at Delhi (77.1°E, 28.7°N) with 300 grid points in the east-west, 170 grid points in the south-
north direction (Fig. 1c), and 50 vertical eta levels with the model top at 50 hPa. The horizontal grid
spacing of the domain is 10 km, while the vertical grid spacing varies from higher resolution (~200 m)
in the boundary layer to coarser resolution (~1200 m) near the model top. We conduct three model
configurations (Table 1) for December 20-24, 2017 to identify the best configuration for
meteorological simulations. The three experiments have been designed with different combinations of

meteorological initial/lateral boundary conditions and planetary boundary layer (PBL) physics. Experiment 1 (EXP1) uses the National Centers for Environmental Predictions (NCEP) Final Analysis (GFS-FNL; 1° x1°, 6 hourly) meteorology data for initial and boundary conditions and the YSU (Yonsei University; (Hong et al., 2006) PBL scheme. Experiments 2 and 3 (EXP2, EXP3) use ERA-Interim Project (1.125° x 0.703°, 6 hourly) for meteorology initial and boundary conditions. EXP2 uses the YSU PBL scheme while EXP3 uses the ACM2 (Asymmetric Convective Model version 2) PBL scheme. ACM2, is a hybrid of the original nonlocal closure (Pleim and Chang, 1992) and a local closure eddy diffusion scheme (Pleim, 2007a, 2007b). The YSU PBL option was coupled with the Noah LSM while ACM2 was coupled with Pleim-Xiu LSM. While YSU permits investigations of both aerosol-radiation (AR) and aerosol-cloud interactions (ACI), ACI is not possible when using the ACM2 PBL scheme because in WRF the ACM2 PBL scheme does not provide the exchange coefficient for heat, which is required to calculate the maximum supersaturations and therefore the activation fraction for aerosol mass and number for each bin/mode, which is based on the Abdul-Razzak and Ghan, (2002) scheme. However, ACM2 has been shown to perform well for air quality in the polluted regions (Mohan and Gupta, 2018; Gunwani and Mohan, 2017; Xie et al., 2012, Mohan and Bhati, 2011). Mohan and Gupta (2018) tested the YSU and ACM2 schemes during the summer time (1-15 June 2010) and focused on the evaluation of temperature, wind speed, PBL height, ozone, and $PM_{10}$. Although these studies recommend using the nonlocal ACM2 PBL scheme for air quality prediction for IGP, there are still seasonal, day-night, and regional biases in the PBL schemes. Gunwani and Mohan (2017) showed that ACM2, QNSE (Quasi Normal Scale Elimination), and MYJ (Mellor-Yamada-Janjić) schemes work well in predicting temperature, humidity, and wind speed in different regions of India. Mohan and Bhati (2011) found that using the ACM2 PBL scheme with Pleim Xiu surface physics improved wintertime meteorology estimates in Delhi indicating its potential in fog predictions, whereas Pithani et al., (2019) recommend using the local PBL scheme MYNN2.5 (Mellor-Yamada-Nakanishi-Niino level 2.5). Shin and Hong (2011) found that a non-local (e.g., ACM2, YSU) scheme is favorable in unstable conditions and a local scheme (e.g., MYJ, Boulac) in stable conditions. All these studies suggest the need for careful consideration of the above-mentioned biases while selecting a PBL scheme. Therefore, to ensure that WRF captures all the relevant meteorological parameters including relative humidity reasonably well during fog events in winter, we designed EXP1, EXP2 and EXP3.

The advantage of Pleim-Xiu LSM (PX-LSM) is that it allows nudging of soil moisture and temperature to improve the prediction of meteorology near the surface (Pleim and Gilliam, 2009; Pleim and Xiu, 2003; Xiu and Pleim, 2001) which Noah LSM does not include. The PX-LSM includes two-layer soil (0–1 and 1–100 cm) model, canopy moisture, and aerodynamic and stomatal resistance. Ground surface (1 cm) temperature is calculated from the surface energy balance using a force-restore algorithm for heat exchange within the soil. Although the two-layer approach in PX-LSM is less detailed than the multilayer soil models such as the Noah LSM (four soil layers; Chen and

Dudhia 2001), it performs well with realistic initialization for soil moisture and through dynamic
adjustment in the model simulation where soil moisture is indirectly nudged according to differences
in 2-m temperature (T2) and 2-m relative humidity (RH) between the model and observation (Pleim
and Xiu, 2003).Soil moisture nudging adjusts the surface evaporation (direct soil surface evaporation,
vegetative evapotranspiration, and evaporation from wet canopies) which then affects the partitioning
of available surface energy into latent and sensible heat flux and thus reduces errors in T2 and 2-m
RH.
For EXP2, meteorological initial conditions were refreshed every 24 hours, while EXP3 was a
continuous run but soil moisture was nudged to the Era-Interim dataset to improve the prediction of
surface fluxes. All other physics and chemistry options are the same for all the experiments except the
surface physics option, which changes with the PBL scheme used. The deposition of cloud droplets is
an important moisture and aerosol sink during fog events. For all these simulations, the deposition
velocity of cloud droplets was reduced to 0.01 m s$^{-1}$ based on Stoke's Law and previous studies
(Katata et al., 2015; Tav et al., 2018) because its default value (0.1 m s$^{-1}$), is large.
To examine the radiative effects of aerosols and aqueous phase chemistry additional
simulations have been done using the meteorological configuration in EXP3, with aerosol-radiation
(wFB) feedback plus aqueous chemistry (wAq.chem), without aerosol-radiation feedback (nFB) but
with aqueous chemistry, and without aqueous chemistry (noAq.chem) but with aerosol-radiation
feedback. The analysis has been done for the fog events on 23$^{rd}$ and 24$^{th}$ December 2017 as

$$F_{ARF}=P(wFB-nFB)$$

$$F_{Aq.chem}= P (wAq.chem-noAq.chem)\_$$

where $F_{ARF}$ is the impact due to aerosol radiation feedback on the meteorological parameters/chemical
species (P) and $F_{Aq.chem}$ is the impact due to the inclusion of aqueous phase chemistry
Emissions used in the WRF-Chem simulations are from the EDGAR-*HTAP v2 (Emissions*
*Database for Global Atmospheric Research- Hemispheric Transport of Air Pollution;0.1° x 0.1°)*
inventory for anthropogenic emissions and *FINN v*2.2 *(Fire INventory from NCAR; 1 km x 1 km)* fire
emission inventory (Wiedinmyer et al., 2011).  Trash-burning emissions (Chaudhary et al., 2021) are
also included in the simulations. The model calculates the biogenic emissions online using MEGAN
v2.04 (*Model of Emissions of Gases and Aerosols from Nature*) (Guenther et al., 2006). The initial and
lateral boundary conditions for chemical constituents are from the global chemistry transport model
CAM-Chem (*Community Atmosphere Model with Chemistry*) (Emmons et al., 2020).
The MOZART (Model for Ozone and Related chemical Tracers) chemical mechanism
(Emmons et al., 2010) is used for gas-phase chemistry, which includes 85 gas-phase species, 39
photolysis, and 157 gas-phase reactions. It has been updated to include an explicit treatment of
aromatic compounds, HONO, $C_2H_2$, and isoprene oxidation scheme (Knote et al., 2014). The lumped
toluene used by Emmons et al., (2010) has been speciated into benzene, toluene, and lumped isomers
of xylenes (Knote et al., 2014). For this study, HCl emissions, transport, dry, and wet deposition are
represented. However, HCl gas-phase reaction is not included in MOZART.
The Model for Simulating Aerosol Interactions and Chemistry (MOSAIC) with four size bins
(0.039–0.156, 0.156–0.625, 0.625–2.500, and 2.5–10.0 µm dry diameters) coupled with MOZART
gas-phase chemistry is used (Fast et al., 2006; Zaveri et al., 2008). The bin sizes are defined by their
lower and upper dry particle diameters, so there is no transfer of particles between bins during water
uptake or loss.  It is assumed that aerosols in each bin are internally mixed with the same chemical
composition while they are externally mixed in different bins.
The aerosol composition includes sulfate ($SO_4^{2-}$), ammonium ($NH_4^-$), nitrate ($NO_3^-$), aerosol
water, sea salt ($Na^+$, $Cl^-$), methanesulfonate ($CH_3SO_3$), carbonate ($CO_3^{2-}$), calcium ($Ca^+$), black carbon
(BC), organic mass (OC), and unspecified inorganic species such as silica, inert minerals, and trace
metals lumped together as other inorganic mass (OIN). For OC, primary OC and secondary OC are
represented separately, where the latter is simulated using the volatility basis set (VBS) approach.
Reactive inorganic species such as potassium ($K^+$) and magnesium ($Mg^+$) are usually present in much
smaller amounts and are equivalent to $Na^+$ since their sulfate, nitrate, and chloride salts are similar in
terms of their solubility in water.
MOSAIC treats condensation and evaporation of trace gases to/from particles, nucleation
(new particle formation), and coagulation. Aerosol coagulation (Brownian) is based on (Jacobson et
al., 1994) and nucleation is based on (Wexler et al., 1994) parameterization of $H_2SO_4$-$H_2O$
homogeneous nucleation. Sulfate, nitrate, chloride, and ammonium aerosols are mainly formed
through oxidation and neutralization/condensation of gas precursors.  Gas-phase sulfuric acid ($H_2SO_4$)
is produced by the gas-phase oxidation of $SO_2$ by OH and nitric acid ($HNO_3$) formation is via the
oxidation of $NO_2$ by OH. HCl is a primary emission product. The neutralization/condensation of
$H_2SO_4$, HCl, and $HNO_3$ with $NH_3$ produces ammonium such as ammonium sulfate (($NH_4$)$_2SO_4$),
ammonium bisulfate ($NH_4HSO_4$), ammonium chloride ($NH_4Cl$) and ammonium nitrate ($NH_4NO_3$),
respectively. The thermodynamic modules in MOSAIC for the dynamic gas-particle partitioning of
aerosols MTEM (Multicomponent Taylor Expansion Method) and MESA (Multicomponent
Equilibrium Solver for Aerosols) calculate the activity coefficient in aqueous phase aerosols and
compute the intraparticle solid-liquid phase equilibrium respectively (Zaveri et al., 2005, 2008). The
Adaptive Step Time-split Euler Method (ASTEM) coupled with MESA-MTEM dynamically
integrates the mass transfer equations.
Aqueous-phase chemistry uses a bulk water approach employing the Fahey and Pandis (2001)
mechanism. It calculates sulfate formation, formaldehyde oxidation, and non-reactive uptake of nitric
acid, hydrochloric acid, ammonia, and other trace gases (Chapman et al., 2009; Pye et al., 2020).
Aqueous-phase sulfate is produced via oxidation of $SO_2$ by $H_2O_2$, $O_3$, TMI (Transition metal Ion:
Fe(III), Mn(II)) catalyzed $O_2$ and $NO_2$. TMI concentrations are prescribed in the model to 0.01 µg m$^{-3}$
for Fe(III) and 0.005 µg m$^{-3}$ for Mn(II) (Martin and Good, 1991). The Fe(III) values are within the
range of water soluble iron in winter time aerosol reported in India (Kumar and Sarin, 2010). Wet
removal (scavenging), is represented by the (Neu and Prather, 2012) scheme for trace gases and Easter
et al., (2004) for aerosols.

**2.2 Observations**

To evaluate the model output, observations of aerosols and meteorology have been obtained
from several satellites as well as ground-based measurement platforms. To examine the aerosol
loading and spatial and temporal distribution, daily Level 2 Aerosol Optical Depth (AOD) retrievals
from the Moderate Resolution Imaging Spectroradiometer (MODIS) aboard Terra and Aqua satellites
are obtained at the spatial resolution of 10 km x 10 km (at nadir) pixel array. It provides aerosol
properties from the Dark Target (DT) algorithm applied over the ocean and dark land (e.g.,
vegetation) and Deep Blue (DB) algorithms over the entire land areas, including both dark and bright
surfaces. Each MOD04_L2 (Terra) / MYD04_L2 (Aqua) products are available at a 5-minute time
interval with an output grid of 135 pixels in width by 203 pixels in length.
The Indian National Satellites (INSAT-3D) in the geostationary orbit at inclinations of 82º
longitude provide an imager fog product (3DIMG_L2C_FOG) with a spatial resolution of 4 km every
30 min (www.mosdac.gov.in). For daytime, the visible channel observation is used to detect fog,
whereas thermal infrared is used to reduce false alarms such as medium/high clouds and snow areas.
INSAT 3D's 'day microphysics' data component analyzes solar reflectance at three wavelengths: 0.5
µm (visible), 1.6 µm (shortwave infrared), and 10.8 µm (thermal infrared). Night-time fog is derived
from TIR-1 (12.0 µm and 10.0 µm) and MIR (10.8 µm and 3.9 µm) channel brightness temperature
over the Indian region. INSAT-3D provides fog intensity varying from 1 to 4 indicating SHALLOW
for visibility > 600 m; MODERATE, DENSE, and VERY_DENSE, respectively for visibility varying
from 0 to 500 m (Banerjee and Padmakumari, 2020). If the visibility is greater than 700 m it indicates
no fog while visibility > 1000 m represents very clear skies. Validation of INSAT-3D fog products
over the IGP shows a 66%-68% probability of detection and a 10% false alarm rate.  It also captures
the entire life cycle of fog from formation to dissipation.  However, detecting fog during multilayer
clouds is still challenging with INSAT-3D (Arun et al., 2018; Chaurasia and Gohil, 2015; Chaurasia
and Jenamani, 2017).
Ground-based monitoring sites provide hourly data of relative humidity, surface temperature,
and wind speed measured by the Central Pollution Control Board, CPCB (http://cpcb.nic.in). Given
the data availability from CPCB stations, nine stations have been considered representing each region
of IGP, which include, Amritsar, IGI Airport (Indira Gandhi International Airport, Delhi), IHBAS
(Institute of Human Behaviour and Allied Sciences, Delhi), Dwarka (Delhi), RKP (Ramakrishna
Puram, Delhi) in the North-West IGP; Kanpur, Lucknow in Central IGP and Patna, Muzaffarpur in
East IGP.

In addition, measurements of several aerosols, trace gases, and meteorology at Delhi (IGI

Airport) from the Winter Fog Experiment (WiFEX) for the period December 10-31, 2017, have also
been used to validate the model output. The WiFEX, an initiative of the Ministry of Earth Sciences
(MoES), India, is a ground-based measurement campaign at the IGI Airport Delhi to understand fog's
physical and chemical features. Additional details of the WiFEX project and related publications can
be found in Ghude et al., (2017).

**3    Meteorology Evaluation**

Previous studies simulating fog highlight the importance of high model vertical resolution

(Pithani et al., 2019; Van Der Velde et al., 2010) for representing the fog formation and the growth of
the fog layer, model initialization (Yadav et al., 2022), initial relative humidity (Bergot and Guedalia,
1994; Pithani et al., 2020), and PBL schemes (Chen et al., 2020; Pithani et al., 2019). In the present
study, 2-m relative humidity (RH2), 2-m temperature (T2), and 10-m wind speed (WS) from WRF-
Chem have been evaluated using ground-based measurements from CPCB monitoring network and
WIFEX campaign for nine stations across the IGP. The comparison of WRF-Chem results with
observations shows that RH2 and T2 are sensitive to the choice of the meteorological initial and
boundary conditions as illustrated by six stations in major cities (Fig. S1). WRF-Chem compares
better with the observations for simulations driven by the ERA-Interim reanalysis than with GFS-FNL
reanalysis since ERA-Interim provides more realistic RH2 than GFS-FNL (Figs. S2 a-f). For example,
RH2 from EXP1 (GFS) varies from 10 to 50%, while RH2 from EXP2 and EXP3 varies from 30 to
100%, which is closer to observation, especially for NWIGP and CIGP. For EIGP, RH2 from EXP1
(GFS) compares better than ERA-Interim, which overestimates the observed RH2. ERA-Interim and
YSU PBL scheme showed damping of RH2 continuously increasing the bias in RH2 with time (not
shown), which was corrected in EXP2 by refreshing meteorology every day at 00h UT during the
model simulation. In addition, maps of surface RH2 and T2 (Figs. S2 g-j) show that the GFS-FNL
dataset has lower relative humidity throughout the domain as compared to ERA-Interim. There are
differences in simulated 2-m temperature between these two datasets which are of smaller relative
magnitude compared to the RH2.
The GFS-FNL driven meteorology EXP1 has a warm bias in NWIGP and CIGP, especially during
night-time, while over EIGP, the model prediction agrees well with observations. EXP2 with the
ERA-Interim driven meteorology and YSU PBL scheme also shows good agreement between
modeled and observed T2 in EIGP. The ERA-Interim driven meteorology with the ACM2 PBL
scheme in EXP3 has a cold bias of up to 7°C over EIGP during daytime from 22nd to 24th December.
The wind speed evaluation shows that WRF-Chem is over-predicting wind speed. However, it is also
possible that some CPCB stations (e.g., Amritsar and RK Puram) have a wind speed low bias due to
the low measurement height and obstructions such as tall trees near the monitoring station as shown in
FigS3. WRF-chem in general overestimates wind speed and several earlier studies have reported this
bias in wind speed (e.g., Mohan and Gupta 2018; Pithani et al.,2019). Moreover, WRF-Chem does not
have the capability to represent building meteorology and parameterizes the effects of urban areas on
meteorology through roughness length, which likely leads to overestimation of wind speed. Note that
at other sites (e.g., over IGI-Delhi and Kanpur) the model measurement agreement is better.
The WRF-Chem performance has been statistically assessed against observation using the Taylor
Diagram (Taylor, 2001), which provides a statistical summary of how well the model output agrees
with the observation in terms of the Pearson correlation, their centered root-mean-square error
(RMSE) difference, and the ratios of their variances (Fig. 2). The centered RMS difference is
proportional to the distance to point "OBS" in the x-axis which measures the extent to which the
simulated and observed datasets match. The centered RMS difference (E'), the correlation (R), and the
standard deviations, $\sigma_m^2$ (simulated) and $\sigma_o^2$ (observed) are calculated as:
$$R = \frac{\frac{1}{N}\sum(M_n - \bar{M})(O_n - \bar{O})}{\sigma_m \sigma_o} \qquad (1)$$
$$E'^2 = \frac{1}{N}\sum[(M_n - \bar{M}) - (O_n - \bar{O})]^2 \qquad (2)$$
$$\sigma_m^2 = \frac{1}{N}\sum(M_n - \bar{M})^2 \qquad (3)$$
$$\sigma_o^2 = \frac{1}{N}\sum(O_n - \bar{O})^2 \qquad (4)$$
where the overall mean of a field is indicated by an overbar.
Each point in the two-dimensional space of the Taylor diagram represents the above mentioned three
different statistical metrics simultaneously, as they are related by the follow equation
$$E'^2 = \sigma_o^2 + \sigma_m^2 - 2\sigma_o \sigma_m R \qquad (5)$$
The diagram is constructed based on the similarity of the above equation and the Law of Cosines:
$$c^2 = a^2 + b^2 - 2ab\cos\phi$$
The percentage bias has also been included to further evaluate the WRF-Chem results. In Fig.
2, better agreement of WRF-Chem results with observations is shown by the marker's proximity to the
"OBS" dashed black line. The WRF-Chem RH has a good correlation for all three experiments with r
> 0.75 at all the locations in IGP for all the experiments. However, the RMSE (shown by red dashed
contours) and the standard deviations are larger for EXP1 compared to EXP2 and EXP3 which lie
closer to the dashed black line indicating that the simulated RH variations are similar to observations.
The mean bias is also large (>20%) for EXP1 (GFS-FNL) for all the stations, marked by red triangles.
For example, simulated RH at Dwarka (4) and Lucknow (7) for EXP2, and IGI Airport (2), IHBAS
(3), Lucknow (7), and Patna (8) for EXP3 show good agreement with observation, with r>0.7,
standard deviation within ±0.25 and mean bias within 10%. Among these stations, the model performs
better for Dwarka (4) and Lucknow (7) for EXP2, IGI Airport (2), and IHBAS (3) for EXP3 with a
smaller centered RMSE (<0.75).
For all the experiments, WRF-Chem T2 agrees well with observations with a correlation
between 0.8 and 0.95. The points are concentrated near the dashed line showing a low RMSE and
standard deviation for T2, signifying a good agreement of simulated T2 with observation in terms of
temporal variation but the T2 relative bias is large for EXP1 (>20%). The RMSE and relative bias for
EXP1 are larger for several of the stations. For example, simulated T2 agrees best with observation at
IHBAS (3) for EXP1 and IGI Airport (2) for EXP2, with smaller centered RMSE and standard
deviation, and bias <5%. The temporal variability of T2 and RH is predicted well for all the
combinations of inputs (Fig. S1), however, the accuracy of simulated T2 and RH is sensitive to the
choice of meteorological initial/boundary conditions. WRF-Chem predicted RH and T2 agree better
with observations when initialized with ERA-Interim meteorology than with GFS-FNL.
The WRF-Chem runs driven by ERA-Interim with YSU (EXP2) and ACM2 PBL (EXP3)
schemes predicted the surface meteorology better over the IGP than the WRF-Chem run driven by
GFS (EXP1). By examining the modeled cloud water content (averaged over all the grids in the
analysis region) in the lowest model level with the INSAT-3D satellite fog intensity for the 23rd and
24th December 2017 (Fig. 3), it is apparent that WRF-Chem with the ACM2 PBL scheme compared
qualitatively well with observations obtained from INSAT-3D satellite in terms of fog coverage over
CIGP, while the WRF-Chem run with the YSU PBL scheme did not produce widespread fog.
However, there is also fog over EIGP in WRF-Chem with the ACM2 PBL scheme although it is not
observed by the satellite. This is because the model has a cold bias in T2 and a high surface RH over
East IGP with ACM2 PBL and Pleim-Xiu surface scheme as discussed earlier, which favors the
formation of fog in this region. The time series in Fig. 4 shows that EXP3 is capable of predicting the
duration of fog on 23rd and 24th December. There is a data gap from INSAT 3D observations because
it is unable to capture fog during daytime in the presence of mid and high-level clouds. Although fog
LWC data is available for Delhi from the WiFEx campaign, the WRF-Chem simulation does not
produce fog in NWIGP, therefore, the WiFEx observations are not included in this evaluation.
In conclusion, EXP3 is the best configuration for predicting fog formation where the ERA-
Interim meteorology, the ACM2 PBL and surface schemes, and soil moisture nudging is used in the
WRF-Chem simulation. Therefore, the evaluation of predicting AOD, surface aerosol concentrations,
and aerosol composition as well as analysis of the impact of aerosols on fog formation uses the EXP3
configuration.

**4 Aerosol Evaluation**
Aerosol is an important factor in correct prediction of fog (Maalick et al., 2016; Stolaki et al.,
2015) as the number of fog droplets depends on the aerosol size distribution and concentration.
Aerosols as CCN can affect the liquid water content in fog and therefore an increase in aerosol
concentration can significantly affect fog lifetime (Stolaki et al., 2015; Zhang et al., 2014b). AOD
retrievals from the MODIS satellite have been used to validate the modeled AOD (Fig. 5). It is
observed that the model captures several important features of the MODIS retrieved AOD spatial
distribution but at the same time somewhat struggles to reproduce the observed AOD magnitude in
some parts of the domain. One possible reason for the underestimation would be the EDGAR-HTAP
emission inventory, which has a low bias for residential sector $PM_{2.5}$ emissions in India (Sharma et al.,
2022).  For instance, the model successfully predicts high aerosol loading seen by MODIS on 20 and
21 December over CIGP and EIGP. This is the region with dense fog both in model and observation.
Higher AOD (>0.5) over CIGP and EIGP can be attributed to the accumulation of aerosols that are
transported by north-westerly winds to these regions from NWIGP (Dey and Di Girolamo, 2011; Jain
et al., 2020; Jethva et al., 2018; Kumar et al., 2018; Yadav et al., 2020). However, WRF-Chem
underestimates AOD over the NWIGP (AOD<0.3) throughout the simulation period and during 23-24
December over CIGP and EIGP where the latter may be related to enhanced scavenging of aerosols by
fog droplets.
The west to east gradient in aerosol loading over IGP is consistent with surface $PM_{2.5}$
distribution (Fig. 6a).  Surface $PM_{2.5}$ concentration is highest in EIGP (>100 µg m$^{-3}$) and it decreases
gradually towards NWIGP (~60-80 µg m$^{-3}$). The time series of $PM_{2.5}$ from CPCB measurements and
the model at stations representative of each region in IGP shows that simulated $PM_{2.5}$ compares well
with observation in terms of day-to-day variation over most of the locations in the IGP (Fig. 6 b-e).
The comparison is good over Amritsar (an NWIGP location), where $PM_{2.5}$ is mostly primary aerosols
from local emissions e.g., residential heating related biomass burning. Agricultural waste burning is at
its peak during post monsoon months (Oct-Nov), whereas during winter burning for residential
heating increases and the stable boundary layer confines these emissions near the surface (Kumar et
al., 2021; Pawar and Sinha, 2022). $PM_{2.5}$ at Amritsar shows a bimodal distribution with morning and
evening peaks whereas it is absent in the model likely due to the absence of diurnal variations in the
WRF-Chem anthropogenic emissions.
At Delhi, the daily variations are predicted well although WRF-Chem underestimates $PM_{2.5}$
observations during the first 4 days. Delhi experiences severe air pollution and haze with high PM
loading (> 500 µg m$^{-3}$) (Bharali et al., 2019). The model is successful in predicting the high $PM_{2.5}$
episode on the 24$^{th}$ of December, but WRF-Chem underpredicts the $SO_4^{2-}$, $NH_4^+$, $NO_3^-$ and $Cl^-$
concentrations (Fig. 7). Although simulated $SO_2$ and $NH_3$ are comparable with observation, sulfate,
and ammonium are underestimated in the model. $SO_4^{2-}$ is underestimated by ~ 9 µg m$^{-3}$, while $NH_4^+$,
$NO_3^-$ and $Cl^-$ are underestimated by ~30 µg m$^{-3}$, ~19 µg m$^{-3}$ and ~40 µg m$^{-3}$ on average, respectively.
In addition, the WRF-Chem model results show that a large percentage of $PM_{2.5}$ is classified as "other
inorganics", which is usually dominated by $PM_{2.5}$ other than BC and OC. The WiFEx observations
show that secondary aerosols contribute ~50% of the $PM_{2.5}$ concentration, which is in line with other
studies that found secondary aerosols contribute 15-50% to $PM_{2.5}$ mass (Sharma and Mandal, 2017;
Behera and Sharma, 2010; Nagar et al., 2017) and $NO_3^-$ constituted 9-13% of $PM_{2.5}$ mass (Lalchandani
et al., 2021; Sharma and Mandal, 2017). This leads to the underestimation of $PM_{2.5}$ over Delhi.
Studies report very high chloride over the IGP with values exceeding 100 µg m$^{-3}$ (Lalchandani et al.,
2021) during winter emitted from increased trash burning and industrial emissions (Pant et al., 2015;
Patil et al., 2013). WRF-Chem incorporates trash-burning emissions which include HCl emissions
from Chaudhury et al.,(2021) for this study however, the inventory contains annual emissions and fails
to resolve the seasonality of trash-burning emissions as identified by Nagpure et al., (2015). They
suggested almost all the waste-burning emissions in neighbourhoods with higher socioeconomic status
in Delhi occur due to the use of waste as cheap heating fuel by individuals such as night watchmen
and pavement dwellers. Chaudhary et al., (2021) considers waste burning that occurs due to lack of
collection infrastructure, and at landfills and, therefore, shows a concentration of waste burning
emissions around the periphery of Delhi but low waste burning emissions in the relatively prosperous
city centre. In addition, emissions from other sources (e.g., industries) are unaccounted for in the
model which likely leads to the underestimation in modeled chloride.
Over the CIGP and EIGP, the underestimation in $PM_{2.5}$ is mostly observed during the dense
fog. It is well known that the hygroscopic aerosols grow in size and are deposited to the surface during
fog (Gupta and Mandariya, 2013; Kaul et al., 2011). $PM_{2.5}$ shows an increase initially with the onset
of fog and then it decreases as the aerosols grow and get deposited through fog droplets. A two order
higher deposition rate (Fig. 6 f, g) during fog compared to the deposition rate of dry aerosol results in
the lower $PM_{2.5}$ over CIGP and EICP during fog events.
A statistical analysis (Table S1) shows a minimum normalized mean bias (NMB) for $PM_{2.5}$ at
Amritsar (2.2%), while in other stations, the normalized mean bias ranges from 48 to 53%, similar to
the reported range of model bias (underestimated by 40–60%) in winter over IGP by earlier studies
(Bran and Srivastava, 2017; Ojha et al., 2020). This was accomplished by incorporating trash-burning
emissions in the model simulation, which improved the $PM_{2.5}$ prediction, increasing NMB by ~4-8%
in IGP. RMSE values range from 41 to 138 µg m$^{-3}$ (normalized RMSE~0.4 to 0.7) comparable to the
reported values by these studies. The Pearson correlation coefficient (r) for the simulated and observed
day-to-day variation in $PM_{2.5}$ lies between 0.4 and 0.7 for all the stations in Fig. 6 except at Patna and
Muzaffarpur which lies within the range in these studies. The poor correlation at Patna and
Muzaffarpur is due to the low modeled $PM_{2.5}$ concentrations which are caused by increased dry
deposition of aerosol particles activated as fog droplets during fog periods, as discussed earlier.
Furthermore, the fog events in WRF and observations have somewhat different time periods causing
WRF-predicted and the observed $PM_{2.5}$ concentrations to decrease at different times.
Previous studies have reported that models tend to underestimate the AOD observation (David
et al., 2018; Pan et al., 2015) during the post-monsoon and winter when agricultural waste burning and
anthropogenic emissions dominate. While anthropogenic emissions include a contribution from the
residential sector, the emissions from small-scale burning for residential heating over IGP especially
during winter are likely underestimated in the current emission inventory (Sharma et al., 2022). This
leads to an underestimation of aerosol concentration in the model. Other possible causes for the
underestimation are the biases in the simulated meteorology (Govardhan et al., 2015; Kumar et al.,
2015; Pan et al., 2015) which affects the aerosol concentration. We corrected some of the biases in
meteorology as discussed earlier however there are still residual biases in the simulated meteorology
e.g., overestimation of wind speed by WRF-Chem. We also observe underestimation of secondary
aerosols over NWIGP which contribute significantly to the aerosol loading over IGP. Secondary
aerosol formation is substantial over CIGP and EIGP in the model compared to NWIGP which will be
discussed in a later section. The underestimation of $PM_{2.5}$ could also be linked to the uncertainty in the
model's chemistry scheme to simulate the secondary aerosols due to missing chemical processes or
due to underestimation of sulfur oxidation at different RH levels (Acharja et al., 2022; Pawar et al.,
2023; Ruan et al., 2022). Moreover, several modeling studies have shown significant improvements in
forecasting surface $PM_{2.5}$ by assimilation of satellite AOD and $PM_{2.5}$ (Ghude et al., 2020; Jena et al.,
2020; Kumar et al., 2020) suggesting the importance of correct initialization of the model in
simulating aerosols over IGP.

**5            Effect of Aerosol Radiation feedback**

Interactions of aerosols with radiation affects temperature and surface heat fluxes, thereby
weakening the turbulence in the PBL and stabilizing the boundary layer height (Fig. 8b) compared to
the clean environment (Fig. 8a). In the presence of well mixed aerosols within the PBL, the radiative
effect of aerosols lowers the noontime PBL height (Fig. 8b). However, the presence of absorbing
aerosols in the PBL warms the air and changes the thermodynamics. Three cases are shown in Fig.
8(c-e) where increases of scattering aerosol concentrations at the top of PBL (Fig. 8c) increases
scattering of radiation by the aerosol layer and reduces the surface reaching solar radiation similar to
Fig. 8b. Higher concentrations of absorbing aerosols at the top of PBL (Fig. 8d) warms the air above
the boundary layer and strengthens the capping inversion stabilizing the PBL and suppressing its
growth. The shallow PBL and weakened daytime vertical mixing confines aerosols and water vapor
near the surface and worsens the air quality of a region. The aerosols trapped in the stagnant PBL
further affects the radiation flux at the surface and creates a positive feedback loop wherein the PBL is
continually suppressed until interrupted by some synoptic weather phenomenon, such as the western
disturbances in the IGP. On the other hand, higher concentration of absorbing aerosols within the PBL
(Fig. 8e) warms the air in the PBL and this results in the higher PBL height. The raised PBL decreases
the aerosol concentration near the surface which is termed as a negative feedback effect.

The aerosol radiation feedback can affect shortwave heating rates (SWHR). The high aerosol
loading over the IGP (Fig. 6 and Fig. 7) allows the AR feedback to reduce the PBL height by more
than 140 m throughout the IGP compared to the surrounding region with AR feedback (Fig. 9a). The
difference in PBL height with and without aerosol radiation feedback is largest during noontime. The
suppressed PBL is due to the decrease in the surface heating flux and the consequent weakening of
turbulence in the PBL. The surface solar radiation flux (SWF) decreases by 5-35 % while the surface
latent heat (LH) and sensible heat (HFX) fluxes decrease by 5-35 % and 10-60 %, respectively (Fig.
S3). The stable, shallow PBL reduces the vertical mixing of aerosols and moisture and confines them
near the surface, resulting in increased $PM_{2.5}$ concentrations and RH near the surface with AR
feedback (Fig. 9). Although T2 should decrease with the reduction in surface SWF, T2 shows mixed
signals with both cooling and warming over IGP. While surface cooling is observed over NWIGP and
EIGP, T2 increases with AR feedback over most of CIGP. The response of AR feedback to T2 varies
in these three regions probably due to differences in the distribution and types of aerosols and the
presence of fog. Increase in surface concentration of $PM_{2.5}$ occurs more over NWIGP and EIGP with
increase in BC and OIN over NWIGP, and sulfate aerosol over EIGP which results in the surface
cooling due to positive AR feedback in these two regions. Over the CIGP, the AR feedback causes a
depletion of surface $PM_{2.5}$ (Fig. 9d), which is likely due to their hygroscopic growth, and then dry
deposition (average $PM_{2.5}$ dry deposition flux of 331 $\mu g \, m^{-2} \, hr^{-1}$ with AR feedback and 282 $\mu g \, m^{-2} \, hr^{-1}$
without AR feedback) in dense fog. The increase in RH with AR feedback favours the growth of
aerosols in size by the uptake of water

Examining further, the time variation of the changes in PBL height, T2, and RH between the

simulations with and without aerosol-radiation feedback (Fig. 9g) shows an increase in T2 while the
surface fluxes, sensible heat flux, latent heat flux, and downward shortwave radiation flux decrease
over CIGP (Fig. 9h). AR feedback affects mostly the lower atmosphere at multiple levels; however,
our finding suggests that the decreased shortwave radiation flux decreases the surface fluxes and thus
the turbulence in the boundary layer resulting in a reduced PBL height on both days. Figure 9 g and h
clearly show a decrease in HFX and LH following the decrease in SWF. Moreover, we observe that
the PBL height is sensitive to latent heat flux likely due to its strong dependence on moisture
availability (Xiu and Pleim, 2001; Zhang and Anthes, 1982).

The impact of AR feedback on T2 depends on factors such as the presence of absorbing

aerosols and their vertical distribution via heating or increased SWF (as observed in CIGP, Fig. S4).
Absorbing aerosols in WRF-Chem include BC and OIN (other inorganic aerosols), which both
increase near the surface (Fig. 9e, Fig. S5) due to their confinement in the stable PBL. Some areas in
the fog-affected region show a decrease in BC as well as $SO_4^{2-}$ likely due to increased dry deposition
in fog water as discussed earlier in this section for $PM_{2.5}$ As a result, AR feedback changes the
absorbing to scattering ratio of aerosols over IGP indicated by the decrease in SSA (Single Scattering
Albedo; Fig. S6). In EIGP, sulfate concentrations are larger with AR feedback than without AR
feedback with time periods where the difference is >1 $\mu g \, m^{-3}$ (Fig.11). The BC concentration changes
are small (<0.5 $\mu g \, m^{-3}$) in the EIGP, resulting in a higher SSA near the surface with AR feedback in
EIGP. In the CIGP, BC concentrations increase while sulfate aerosols decrease within the PBL with
AR feedback (Fig.11) compared to the simulation without AR feedback. A decrease in SSA is seen for
the CIGP throughout the boundary layer while in EIGP the decrease occurs near the top of the PBL;
difference in SSA due to AR feedback is negligible in NWIGP. Also contributing to the higher SSA in
EIGP is the increase in RH (Fig. 9) due to AR feedback favoring the growth of aerosols in size by
uptake of water and the production of secondary aerosols such as $SO_4^{2-}$ and $NH_4^+$.
A similar observation has been made by Ramachandran et al., (2020) where SSA decreases
with increasing altitude due to absorbing carbonaceous aerosols at higher elevations which contributes
$\geq 75\%$ to the aerosol absorption over IGP. Increased shortwave heating (Fig. 10) is probably caused by
the increased absorbing aerosols near the surface which overwhelms the surface cooling due to
reduced shortwave radiation at the surface.
The increase in 2-m RH is substantial over CIGP on 24[th] December (Fig. 9g) compared to the
previous day following the decrease in PBL height which constrains the moisture near the surface.
The decrease in RH by 2% or more when aerosol-radiation feedback is included compared to no
aerosol-radiation feedback is likely due to increase in T2. However, the increase in RH in the
afternoon associated with a decrease in LH and PBL height is important for the air to saturate which
then favors the formation of fog in a polluted environment. Note that the increase in T2 with AR
feedback is very small ($<0.5°C$) which reduces further after noon (~12:30 pm IST) on both days.
Another important factor that can affect the extent of change in PBL height is the distribution
of aerosols in the vertical (illustrated in Fig. 8). The pressure-time cross-sections of differences in T,
$PM_{2.5}$, BC, and $SO_4^{2-}$ between aerosol radiation (AR) feedback (wFB) and no aerosol radiation
feedback (nFB) for three regions, NWIGP, CIGP, and EIGP are shown in Fig. 11. The difference in
the PBL height reaches a maximum with the AR feedback during midday (12:30-15:30 IST). Increase
in temperature in the boundary layer is observed with AR feedback particularly at the upper PBL in all
the regions of IGP. This induces a temperature inversion resulting in a stable and suppressed PBL. In
all the regions the decrease in PBL height (100-200 m) is larger on 24[th] December compared to 23[rd]
December. The difference in the PBL height on 23[rd] and 24[th] December with AR feedback on these
days is possibly controlled by the aerosol distribution during the previous day or early morning on the
same day. For example, in all the regions an increase in $PM_{2.5}$ is observed the previous night (23:30
onwards) till ~11:30 of December 24, with increased BC over NWIGP and CIGP whereas both BC
and $SO_4^{2-}$ over EIGP. The increased $PM_{2.5}$ concentrations suppress the development of the PBL after
sunrise with AR feedback on December 24 compared to that on December 23, leading to the observed
differences in $\Delta$PBL height on these two days. Increase in BC concentrations in NWIGP and CIGP are
found above the PBL on 24[th] December whereas BC concentrations decrease within the PBL. This BC
concentration gradient creates a temperature inversion, for example between 10:30-14:30 IST. The
increase in BC warms the air in the PBL; however, the warming is not strong enough to cause
negative feedback over CIGP. On 23[rd] December a small increase in BC is uniform throughout the
PBL, while there is a decrease in $SO_4^{2-}$ concentrations, resulting in a warmer PBL (Fig. 11) with AR
feedback.
In EIGP, BC distribution is similar to that in CIGP with AR feedback while there is a
substantial increase in sulfate aerosol in the PBL. This results in the strongest extinction in EIGP as
evident from the largest difference in PBL height and surface cooling with AR feedback among the
three regions.  Although $\Delta$PBL is small on 23[rd] December, it still results in the accumulation of
aerosols during night-time (~23:30 pm onwards) which further strengthens the AR feedback effect the
next day in NWIGP and CIGP. Thus, AR feedback stabilizes the PBL, increases $PM_{2.5}$ and RH in the
PBL making conditions favourable for persistence of fog over IGP.

**6        Effect of Aqueous phase chemistry**

In this section we discuss the impact of aqueous phase chemistry on aerosol composition and

its interaction with meteorology. There is a considerable difference in the surface concentration of
$PM_{2.5}$ (>16 µg m$^{-3}$) in the absence of aqueous chemistry over CIGP and EIGP where fog occurs (Fig.
12a) while the difference is negligible over NWIGP where fog does not occur. This is due to the
formation of secondary aerosols through aqueous phase chemistry and the hygroscopic growth of
aerosols during fog in these regions with the inclusion of aqueous chemistry in the model. In the
region between CIGP and EIGP (83E-84E; marked by the box in Fig. 12a), $PM_{2.5}$ concentration is less
in the simulation with aqueous-phase chemistry than without aqueous-phase chemistry because
deposition of fog water aerosols to the surface increases as the fog thickens (Fig. 13, Fig. S7). Figure
13 shows the relation between formation of secondary aerosols, deposition flux of $PM_{2.5}$, and fog with
and without aqueous phase chemistry. During the fog event, the secondary aerosols ($SO_4^{2-}$ , $NH_4^+$)
increase significantly by 4-10 µg m$^{-3}$ due to aqueous phase chemistry adding to the $PM_{2.5}$ burden over
IGP. The intensity of fog is high around midnight December 24-25 compared to that on 23$^{rd}$ and 24$^{th}$
(1:30-11:30 IST)) which increases the dry deposition flux of $PM_{2.5}$ causing a sharp drop in the $PM_{2.5}$
concentration on 24$^{th}$ December compared to the previous night's fog event. The observed change in
$PM_{2.5}$ over a region is the net result of the formation of secondary aerosols and its deposition with fog
droplets.

The composition distribution of $PM_{2.5}$ (Fig. 12b) has a similar distribution for the simulations

with and without aqueous phase chemistry over NWIGP where fog did not occur. The primary
aerosols are higher (BC > 9%, OC ~ 16-30%, OIN > 50%), than the secondary aerosols (<5%). While
the model requires fog for accelerated formation of secondary inorganic aerosol, experimental data
(Fig. 7) supports significant formation of secondary inorganic aerosol at elevated RH levels even in
haze aerosol (Acharja et al., 2022). On the other hand, the central and east IGP stations are fog-
covered and therefore, there is an increase in secondary aerosols especially $SO_4^{2-}$ and $NH_4^+$ when
aqueous phase chemistry is included in the simulation. $SO_4^{2-}$ is chemically produced via aqueous
phase chemistry in cloud water, hence the abrupt increase whereas $NH_4^+$ maintains a gas-aerosol and
gas-cloud equilibrium with $NH_3$ and $SO_4^{2-}$ via neutralizing the drop or aerosol. $NO_3^-$ is high in the
model compared to $SO_4^{2-}$ and $NH_4^+$ and it decreases by ~1-2 % with aqueous phase chemistry. We
observe a small increase in $NO_3^-$ during fog, however it drops as fog intensifies, more rapidly than that
without aqueous phase chemistry likely due to increase in dry deposition. This results in lower
average $NO_3^-$ to $PM_{2.5}$ ratio with aqueous phase chemistry. Moreover, $NO_3^-$ is high over the fog
covered CIGP and EIGP compared to NWIGP suggesting that transport and chemistry of $NO_x$ in
CIGP and EIGP produce more $HNO_3$. Aerosol $NO_3^-$ is also in equilibrium with $HNO_3$ and it is formed
only if excess $NH_3$ is available beyond the sulfate neutralization. Thus, $NH_4^+$ and $NO_3^-$ changes are
likely due to changing the partitioning between gas and liquid based on the production of sulfate.

$PM_{2.5}$ is mostly composed of organic aerosols (OA) over CIGP and EIGP (Lalchandani et al.,

2021; Srinivas and Sarin, 2014) whereas $PM_{2.5}$ is OIN (dust) and OA over NWIGP (Ram et al., 2012a;
Sharma and Mandal 2023). Although observational studies report $Cl^-$ as one of the largest contributors
(12-17%) to $PM_{2.5}$ after the organics (Lalchandani et al., 2021; Pant et al., 2015) during winter, $Cl^-$ is
largely underestimated by the model as discussed in section 4 and contributes only ~3%. A small
increase (2-4%) in secondary organic aerosols (SOA) from glyoxal production in aerosols occurs for
the simulation with aqueous phase chemistry included during intense fog, suggesting there are
feedbacks between cloud chemistry (without glyoxal aqueous chemistry) and aerosol chemistry.
However, similar to $NO_3^-$, average SOA (ASOA (anthropogenic)+BSOA (biogenic) + GlySOA)
shows a decrease when aqueous phase chemistry is included. SOA contributes significantly to organic
aerosol loading over IGP (Kaul et al., 2011; Mandariya et al., 2019).

The WRF-Chem results on aerosol composition during fog behave similarly to those of

previous observational studies. For example, Ram et al., (2012a) reported an increase of EC, OC, and
WSOC concentrations by ~30% during fog and haze events at Allahabad, a location in the Central
IGP, and a marginal increase of these constituents at Hisar (NWIGP). Several studies report an
increase in inorganic ions ($NH_4^+$, $NO_3^-$, and $SO_4^{2-}$) during fog over IGP and elsewhere (Gundel et al.,
1994; Ram et al., 2012a). Recent studies suggest that a significant fraction of atmospheric particulate
matter in the IGP is comprised of carbonaceous aerosol (~30–35% of the PM) and water-soluble
inorganic species (~10–20% of the PM) during October−January when emissions from biomass
burning (including residential heating) are dominant over IGP (Ram et al.,2012b; Rengarajan et al.,
2007; Tare et al., 2006).

Both the simulations with and without aqueous-phase chemistry include the AR feedback. The

aqueous chemistry increases the mass of $PM_{2.5}$ and the size of the aerosols, both of which contribute to
AR feedback, thus increasing RH and PBL stability. The increase in RH also saturates the air,
promotes aerosol growth by water uptake, and thus favors fog formation. Since the secondary
inorganic aerosols are scattering aerosols, the increased scattering of radiation further reduces the
solar radiation reaching the surface (Fig. 14a). Over CIGP the presence of higher aerosol loading
reduces the T2 during daytime, particularly on the 24[th] of December which then reduces the PBL
height and increases RH near the surface (Fig. 14b). These conditions favor fog formation over the
CIGP. Further, the fog water content with aqueous-phase chemistry is higher than that without
aqueous-phase chemistry on 24[th] December post-midnight (Fig. 13b). This is likely due to saturation
of air due to increase in RH and lower T2, induced by the AR feedback caused by the increase in
$PM_{2.5}$. Although the difference in T2 is small (<0.4), favourable conditions mentioned above are
conducive to fog formation. Because aqueous chemistry increases sulfate concentrations, the size of

the aerosols also increase. The increased aerosol size, which can grow further by water uptake, also impacts the solar radiation reaching the surface, affecting fog formation and dissipation.

### 7    Effect of AR feedback and aqueous chemistry on the duration of fog

Aerosol and its radiative effects impact fog characteristics, including the fog liquid water content (LWC), the fog lifetime over a region and hence its spatial and temporal distribution. Variations of fog LWC in WRF-Chem contrast the fog in the CIGP and EIGP (Figure 15) as well as among the three experiments (with aqueous chemistry plus AR feedback, with aqueous chemistry without AR feedback, and without aqueous chemistry but with AR feedback). WRF-Chem does not simulate fog over NWIGP in the model for the study period. In Figure 15, only foggy grid points are considered for the first fog event on 23-24 December. The LWC is 5-15% higher with AR feedback than without AR feedback and without aqueous phase chemistry for both CIGP and EIGP. The interquartile range is larger for the simulation with and without AR feedback than without aqueous phase chemistry in CIGP showing large variability in the LWC.  On the other hand, in EIGP the variability in LWC is greater in the simulation with AR feedback compared to the other two experiments.

The formation and dissipation times of the two fog events for the three experiments are listed in Tables 2 and 3 for CIGP and EIGP. The 23-24 December fog starts forming two hours earlier and the 24-25 December fog forms one hour earlier in both CIGP and EIGP with AR feedback than without AR feedback. In the simulation without aqueous phase chemistry, fog formation is delayed by an hour or two compared to the simulation with aqueous chemistry plus AR feedback in CIGP. In EIGP the 23-24 December fog forms at the same time with AR feedback and without aqueous phase chemistry while the 24-25 December fog is delayed by an hour without aqueous phase chemistry. Fog dissipation usually occurs after sunrise when the shortwave radiative warming at the surface warms the air, which results in PBL mixing. In addition, absorbing aerosols like BC affect fog dissipation by increasing the radiative heating in and above the fog. We find an increase in BC and shortwave heating in the PBL with AR feedback (Fig. 10,11) and warming over CIGP with AR feedback. Fog intensity starts to decrease after 01:00 UTC (06:30 IST), however, in our study, we find that the fog dissipates completely in the afternoon (~10:00 UTC or 15:30 IST) for both the simulations with AR feedback and no aqueous chemistry while an hour later without AR feedback in CIGP. Fog dissipation is delayed in EIGP with AR feedback compared to that without AR feedback and without aqueous phase chemistry. In both the regions, fog lifetime increases with AR feedback. All the stations, however do not show the same pattern, for example, the 23-24 December fog in Lucknow forms and dissipates at the same time for simulations with AR feedback and without aqueous phase chemistry, and the 24-25 December fog forms later with AR feedback than without AR feedback. Patna shows no difference in the 24-25 December fog formation in all the three experiments. To gain better insights on the fog timing, we recommend that simulations at higher spatial and temporal resolutions be performed to represent better the fog dynamics at point locations. Furthermore, there are other

important factors to consider, e.g., improved emissions, better simulations of aerosol chemical
composition, and evaluation of aerosol deposition.
The AR feedback and aqueous-phase chemistry have the potential to impact aerosol-fog
interactions. We can learn about the effect of the aerosol-radiation interactions on CCN concentrations
because the WRF-Chem model calculates the CCN concentrations at different supersaturations as a
diagnostic output. We compare CCN at 0.02% supersaturations, a value typical of fog, among the
three experiments. For the 23-24 December fog in CIGP, hourly CCN concentrations are ~10% higher
for the simulations with AR feedback with or without aqueous chemistry than with no AR feedback
(Figure S8) during the first 8 hours of the fog event (16:00-24:00 IST 23 December). Surprisingly, the
simulation with no aqueous chemistry has higher CCN concentrations than the simulation with
aqueous chemistry, as more CCN are expected with aqueous chemistry, because aqueous chemistry
adds sulfate to the aerosol mass increasing the mass of $PM_{2.5}$. Increased $PM_{2.5}$ further contributes to
AR feedback, increasing RH which favours the growth of the aerosol size, categorizing more aerosols
as CCN. However, the dry deposition flux (ddmass) also increases in dense fog which causes rapid
loss in CCN and activated aerosols during fog events with the AR feedback (Fig. S7) and more so
without aqueous-phase chemistry. Shao et al. (2023) examined aerosol-fog interactions for two
consecutive fog events by comparing WRF-Chem results with current emissions strengths to those
with low emission strengths. They show that the first fog event promotes formation of the second fog
event leading to wider fog distribution, and longer fog lifetime favoured by multiple feedbacks
including AR feedback i.e., low temperature, high humidity and high stability similar to our study.
While Shao et al. (2023) observe a delay in dissipation of the first event and early formation of second
fog event, we find an early dissipation and early formation of fog with AR feedback as discussed
earlier in this section. In summary, aqueous phase chemistry together with AR feedback promotes
early formation of fog while AR feedback alone promotes early dissipation of fog and plays a critical
role in the formation and evolution of the fog over IGP.

**8   Conclusions**
The effects of aerosol-radiation (AR) feedback and aqueous chemistry in air quality and fog
have been assessed over IGP.  We carried out three experiments using WRF-Chem testing different
combinations of PBL schemes and meteorology initial and boundary conditions. The best
representation of surface meteorology for the IGP region for the case study (December 20-24, 2017)
used ERA-Interim reanalysis to drive the meteorology and ACM2 PBL scheme with soil moisture
nudging to ERA-Interim. With this meteorology configuration for WRF-Chem, evaluation of aerosol
concentrations with measurements and the impact of aerosols on atmospheric processes during fog
were examined. Further, we included trash-burning emissions to represent anthropogenic chloride
aerosols in our configuration. Incorporation of trash burning emissions did improve the model
simulations of $PM_{2.5}$ and better captured the day-to-day variability of $PM_{2.5}$ in IGP, yet underestimated

its magnitude compared to CPCB observations. Moreover, secondary aerosols particularly, chloride aerosols are underestimated in the model. This underestimation is likely caused by a low bias in the residential burning emission inventory and a failure of the emission inventory to represent residential sector emissions from the use of trash as cheap heating fuel properly. AOD regional distribution is predicted well by the model for most of the IGP. However, AOD is underestimated over NWIGP likely due to an underestimation of fugitive emissions during wintertime cold spells.

The AR interactions showed a significant impact on meteorology and air quality over IGP. A WRF-chem simulation with AR interactions resulted in a lower PBL height by ~50-270 m compared to a simulation without AR interactions leading to accumulation of aerosols and moisture near the surface. Reduced surface shortwave radiation flux and the surface sensible and latent heat fluxes due to aerosol radiative effect suppressed the turbulence resulting in a stable PBL. The shallow PBL further increased surface $PM_{2.5}$ (> 8 µg m$^{-3}$) and RH (2-8%) over IGP and this positive feedback mechanism promoted thickening of fog over IGP. However, an increase in absorbing aerosols in the PBL gave negative feedback, increasing the shortwave heating and temperature particularly over CIGP. Fog forms when air is saturated which occurs when the surface temperature is reduced or the moisture content increases causing saturation of air. This study suggests that increase in RH saturated the air and the increase in aerosols favoured fog formation as depicted by the thickening of fog intensity. Aqueous phase chemistry on the other hand contributed significantly to secondary aerosols in the fog, especially sulfate aerosols, indicating substantial formation of secondary aerosols in the cloud. The underpredicted secondary aerosols over NWIGP where no fog occurred implies underestimation of formation of aerosols through gas and aerosol chemistry in the model. This underestimation could also be linked to an underestimation of pH in the default MOSAIC scheme (Ruan et al., 2022) which slows the secondary aerosol formation, or an underestimation of the aqueous sulfur oxidation in haze aerosol at > 80% RH before the onset of fog (Acharja et al., 2022), or missing multiphase oxidation processes (Wang et al., 2022). Nevertheless, we find that the model successfully simulates the same changes in the inorganic composition during fog in IGP as reported by observational studies referred earlier in section 6. We also observed that AR feedback with aqueous chemistry initiated the fog formation 1-2 hours earlier than the initiation time in the simulation without AR feedback and without aqueous phase chemistry whereas AR feedback alone led to early dissipation of fog. In addition, fog acted as an important sink of aerosols in a polluted environment with increased dry deposition with cloud water. Thus, AR feedback and aqueous chemistry play a significant role in modulating the distribution and concentration of aerosols and evolution of fog in the PBL. Aerosol-cloud interactions were not investigated in this study due to the limitation of the ACM2 PBL scheme in providing necessary information with other modules in WRF. Previous studies of aerosol-fog interactions have found that ACI also promotes early onset of fog formation and increases fog duration (Maalick et al., 2016; Yan et al.,2021). While these previous studies were applied to

midlatitude fog events, it is likely that ACI also plays a dominant role in IGP fogs, suggesting that
future studies are needed to fully understand aerosol effects on IGP fog events.
The large emission of aerosols and trace gases in the IGP makes the atmospheric dynamics as
well as chemistry complex, suggesting the need for more studies using both models and ground-based
measurements to better understand the processes. While all aerosol types interact with solar radiation
and reduce the surface reaching flux, presence of absorbing aerosols in the boundary layer and its
vertical distribution plays an important role in modulating the meteorology over IGP. It is therefore
crucial to improve the simulation of absorbing aerosols e.g., BC in the vertical as well as at the surface
to increase the accuracy in predicting formation as well as the dissipation of fog in this region.
Emissions from burning for residential heating are an important source of aerosols in IGP during post-
monsoon and winter and the inclusion of these sources in the emission inventory would improve the
prediction of wintertime aerosols. For example, the underestimation of chloride aerosol in the model
indicates unaccounted emission sources over IGP and the need for more work on better quantifying
trash burning emissions, which may not only improve particulate chloride in the model but also
improve simulations of other aerosol chemical components through aerosol thermodynamics.
Additionally, more detailed modeling studies are required to understand the missing chemical
processes if any in the model which leads to biases in sulfate, nitrate and ammonium partitioning
between gas and aerosol phases. We find that the change in PBL height with AR feedback is sensitive
to changes in LH, signifying the role of soil moisture in PBL dynamics. Several studies have reported
cooling over IGP due to an increase in irrigation (Kumar et al., 2017; Mishra et al., 2020). Further
investigations into the role of irrigation in the increasing fog events over NWIGP would help in better
understanding the formation and persistence of fog over this region. It can be concluded that fog
forecasting is a complex process due to the multiple factors involved and this work suggests that AR
feedback is important in fog forecasting while aqueous phase chemistry plays an important role in
defining the composition of aerosols over IGP.

**Acknowledgement**
This material is based upon work supported by the NSF National Center for Atmospheric Research
(NCAR), which is a major facility sponsored by the U.S. National Science Foundation under
Cooperative Agreement No. 1852977. CB is thankful to the Fulbright Kalam Climate Fellowship
program under USIEF (United States – India Educational Foundation), and Women Scientist (WOS-
A) program, Department of Science and Technology (DST), Govt of India. The authors acknowledge
the use of MODIS data from NASA's Land, Atmosphere Near real-time Capability for EOS (LANCE)
system (https://earthdata.nasa.gov/lance ), part of NASA's Earth Observing System Data and
Information System (EOSDIS); Meteorological & Oceanographic Satellite Data Archival Centre
(MOSDAC: https://www.mosdac.gov.in/), Space Applications Centre, Indian Space Research

Organisation, Govt. of India for INSAT-3D fog data and the Central Pollution Board of India (CPCB: https://app.cpcbccr.com/ccr/#/login ) for meteorology data. We would like to acknowledge the high-performance computing support from Cheyenne (doi:10.5065/D6RX99HX) provided by NCAR's Computational and Information Systems Laboratory. We thank Duseong Jo and Behrooz R, and the two anonymous reviewers for their constructive comments on the manuscript.

**Data availability:** All the model simulations are archived on the NCAR campaign storage (/glade/campaign/acom/acom-weather/chandrakala) and can be accessed by contacting the corresponding author. WIFEX data can be made available by contacting Dr S.D. Ghude. Trash Burning emission data is available on Mendeley data (doi- http://dx.doi.org/10.17632/t2tn4t9473.1). MODIS AOD retrievals can be downloaded from https://earthdata.nasa. gov/. Meteorology and aerosol data from the Central Pollution Control Board (CPCB) is available at http://cpcb.nic.in.

**Author contributions:**

CB: Conceptualization, Formal Analysis, Writing

MB: Conceptualization, Supervision, Writing-review and editing, Funding acquisition

RK: Conceptualization, Supervision, Writing-review and editing

SDG: provided ground-based observation data, Writing-review and editing

VS and BS: provided trash burning emission data, Writing-review and editing

**Competing interests:** The authors declare that they have no conflict of interest.

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

Table 1 Experiment set-up for the study. Numbers in parentheses for the physics options denote the
namelist settings of the WRF-Chem model.

| | EXP 1 | EXP 2 | EXP 3 |
|---|---|---|---|
| Meteorology Initial /lateral boundary Condition: | *NCEP Final Analysis (GFS-FNL), 1° x1°, 6 hourly* | *ERA-Interim Project, 1.125° x 0.703°, 6 hourly* | *ERA-Interim Project, 1.125° x 0.703°, 6 hourly* |
| Physics Options | | | |
| Cloud Physics | *Morrison 2-mom (10)* | *Morrison 2- mom (10)* | *Morrison 2- mom (10)* |
| Longwave Radiation | *RRTMG scheme (4)* | *RRTMG scheme (4)* | *RRTMG scheme (4)* |
| Shortwave Radiation | *Goddard shortwave (2)* | *RRTMG scheme (4)* | *RRTMG scheme (4)* |
| Surface Layer Physics | *Revised MM5 Monin-Obukhov scheme (1)* | *Revised MM5 Monin-Obukhov scheme (1)* | *Pleim-Xiu (7)* |
| Surface Model | *unified Noah land-surface model (2)* | *NoahMP (4)* | *Pleim-Xiu (7)* |
| PBL Scheme | *YSU scheme (1)* | *YSU (1)* | *ACM2 (7)* |
| Convective Parameterization | *Grell-Freitas (3)* | *Grell-Freitas (3)* | *Grell-Freitas (3)* |
| | *Continuous simulation* | *\*Meteorology refreshed every 24 hr* | *\*\*Continuous simulation: Soil nudging included* |




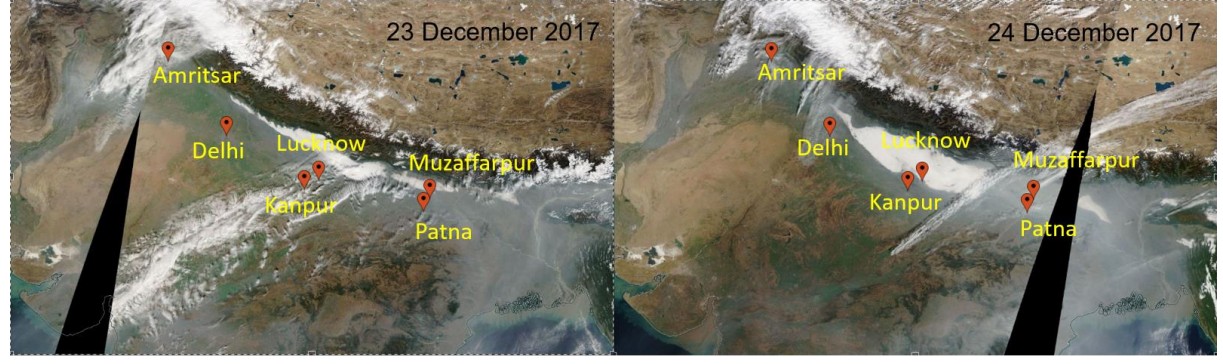

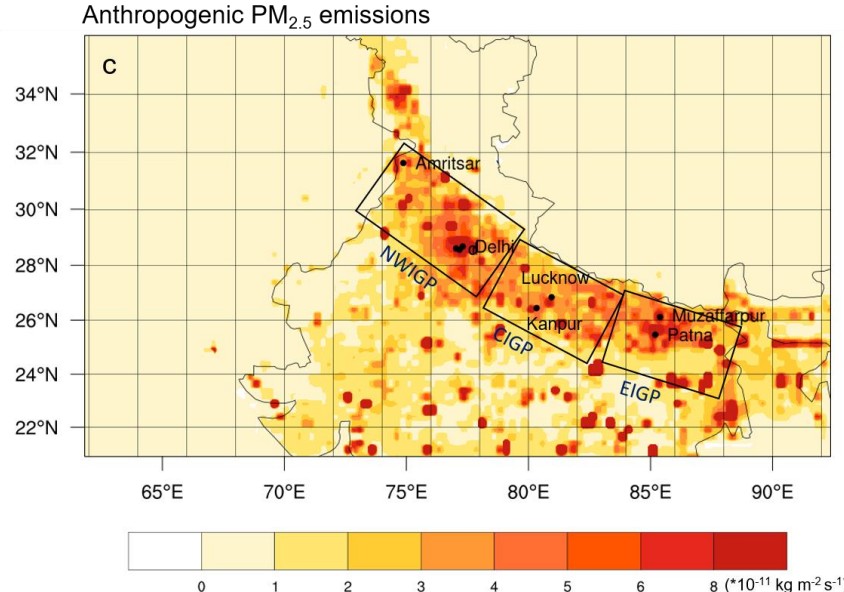

Figure 1 The MODIS reflectance (true color) map representing low cloud over Indo Gangetic Plains, India (study region) indicative of likely fog and haze on 23rd December (a) and 24th December (b) 2017. (c) Anthropogenic emission of $PM_{2.5}$ over IGP for December 2017 obtained from EDGAR-HTAP. The boxes represent the regions Northwest IGP (NWIGP), Central IGP (CIGP), and East IGP (EIGP).

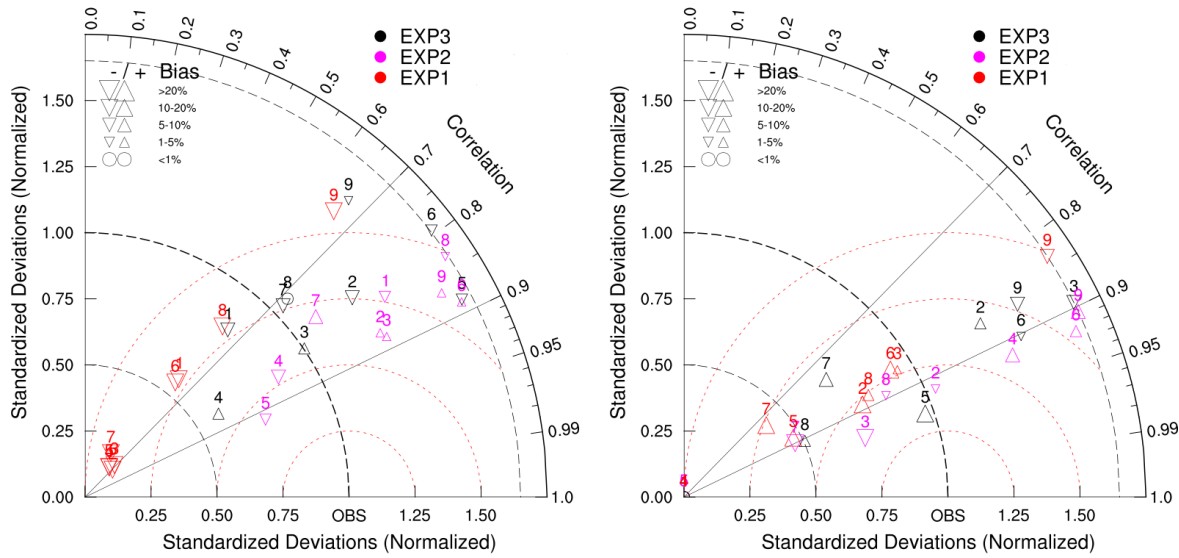


Figure 2 Taylor Diagram of simulated (WRF-Chem) and observed (CPCB) relative humidity (left) and
2-m temperature (right) over IGP. The colors indicate the experiments. The red dotted contours
represent RMS values. The marker (triangles) size varies with a mean bias between the experiments
and observation. Upside-down triangles represent positive bias (exp-obs) and vice versa. The stations
over IGP are denoted by number 1. Amritsar, 2. IGI Airport (Delhi), 3. IHBAS (Delhi), 4. Dwarka
(Delhi), 5. RKP (Delhi), 6. Kanpur, 7. Lucknow, 8. Patna, 9. Muzaffarpur. The locations are marked
in Fig.1a.

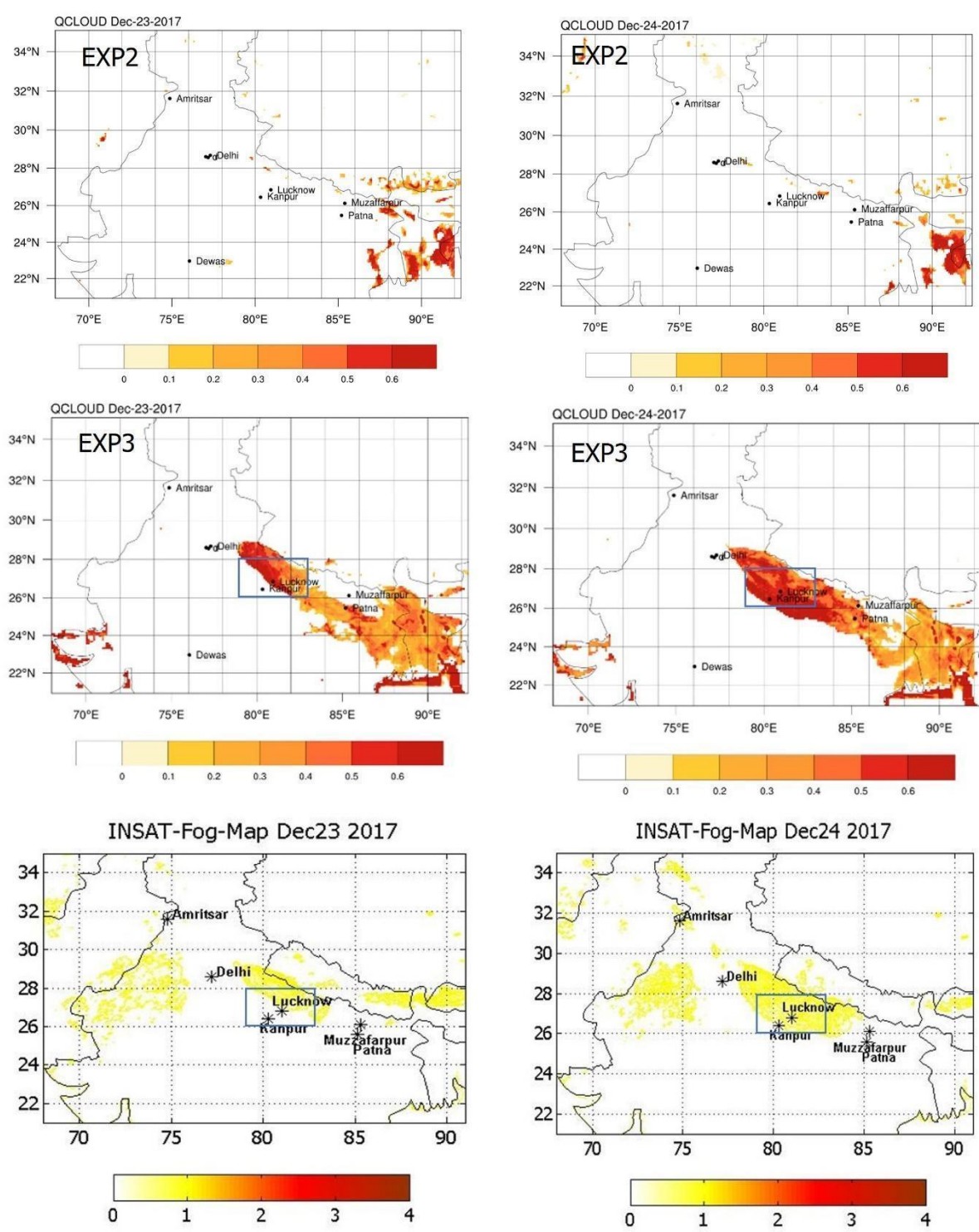


Figure 3 Average WRF-Chem surface layer cloud water mixing ratios (g m⁻³) for EXP2 and EXP3

(top four panels) and INSAT-3D satellite fog intensity (bottom two panels) on 23 and 24 December

2017. INSAT-3D satellite fog intensity varies from 0 to 4 indicating SHALLOW, MODERATE,

DENSE, and VERY_DENSE, respectively. The rectangle in central IGP is the region for the time

series analysis.


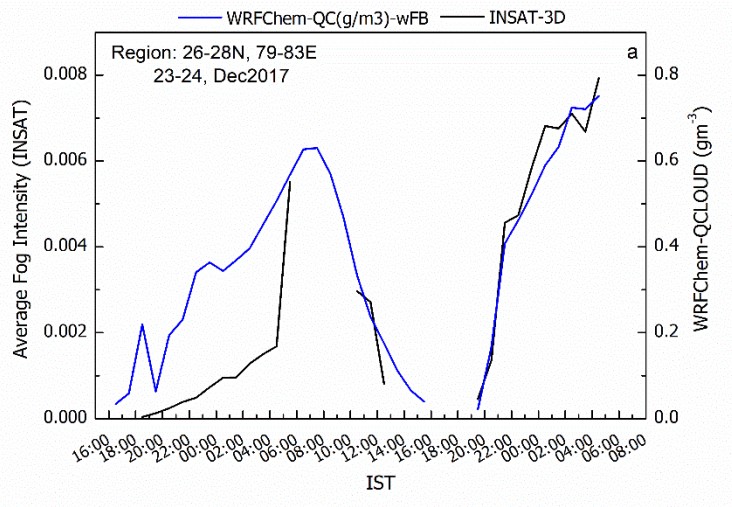


Figure 4 Average Hourly variation of fog on 23 and 24 December 2017 from WRF-Chem EXP3

simulation and INSAT-3D satellite between 26°N-28°N,79°E-83°E (region shown in Fig 3). The time

is in IST (Indian Standard Time; IST is 5.5 hours ahead of Universal Time Coordinate (UTC).



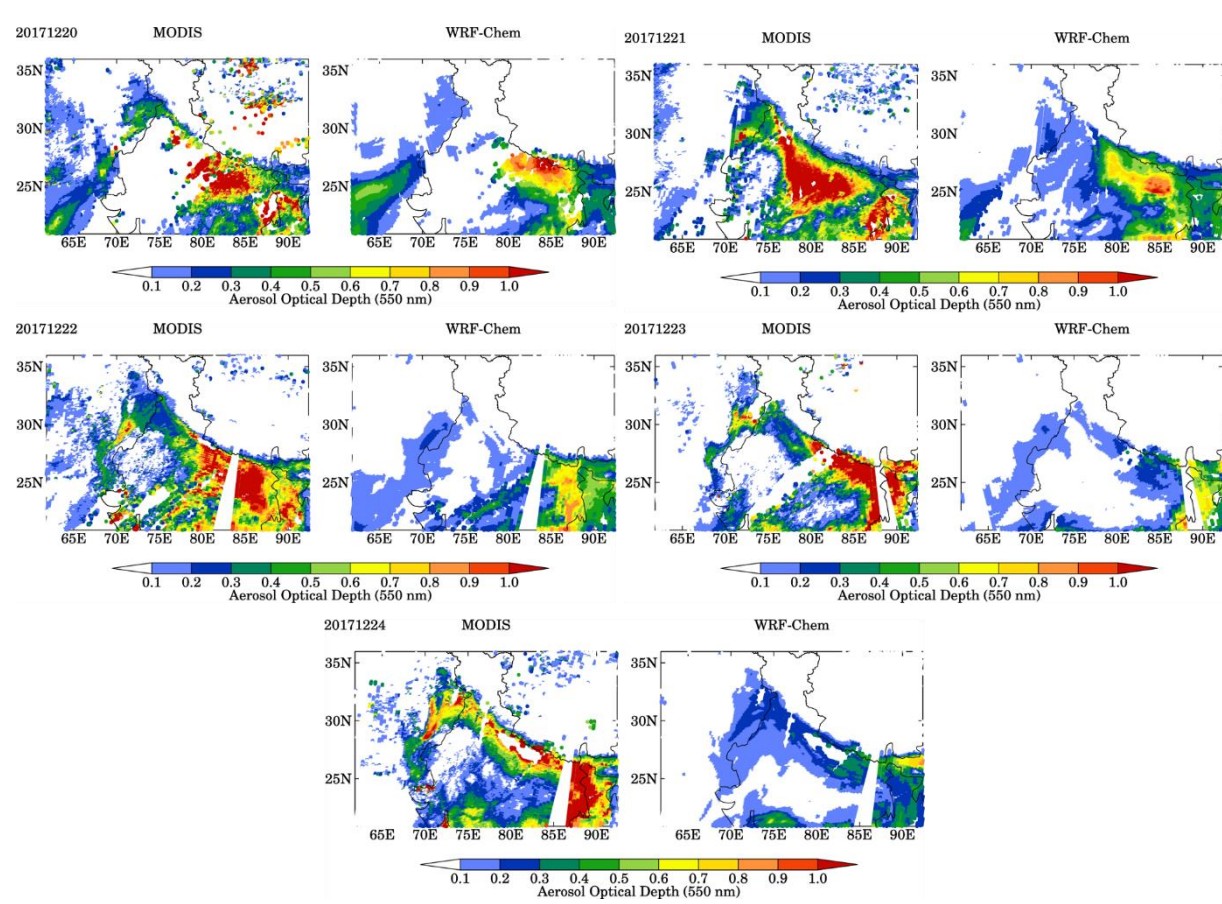


Figure 5 Comparison of WRF-Chem AOD with MODIS observation over the model domain on 20,
21, 22, 23, and 24 December 2017.


1310          .


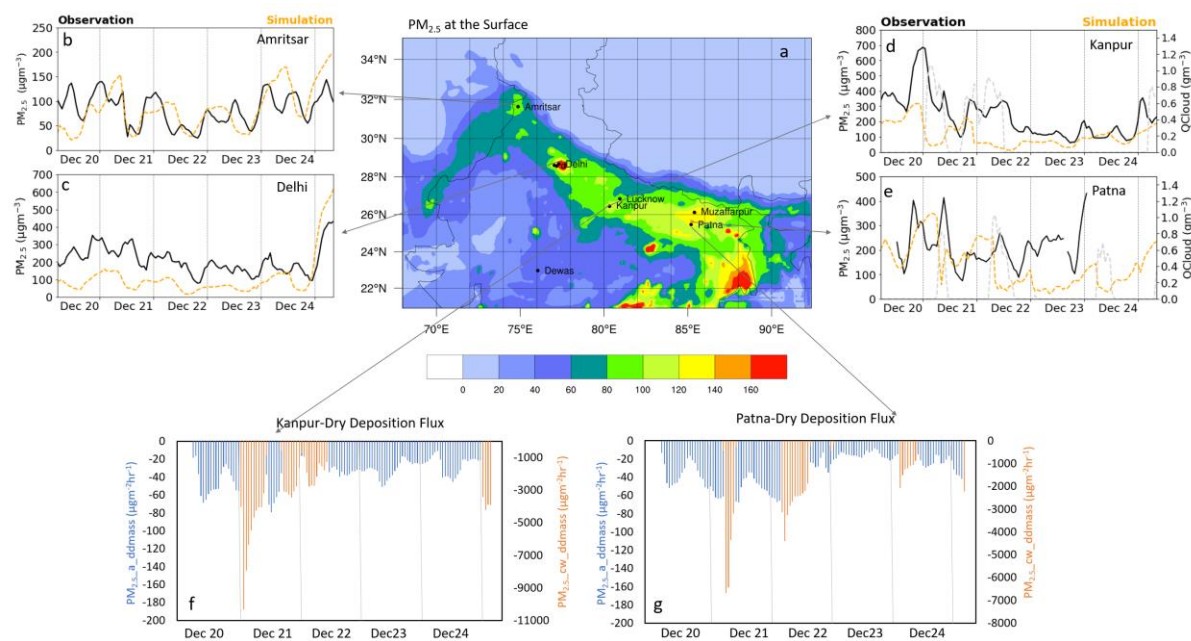


Figure 6 WRF-Chem simulated surface PM$_{2.5}$ map over IGP (a); comparison of WRF-Chem PM$_{2.5}$
with CPCB observation for the period 20-24 Dec 2017 for (b) Amritsar, (c) Delhi, (d) Kanpur and (e)
Patna. Dry Deposition rate of PM$_{2.5}$ for (f) Kanpur and (g) Patna. The grey dotted line in (d) Kanpur
and (e) Patna is fog (QCloud) present during the study period.

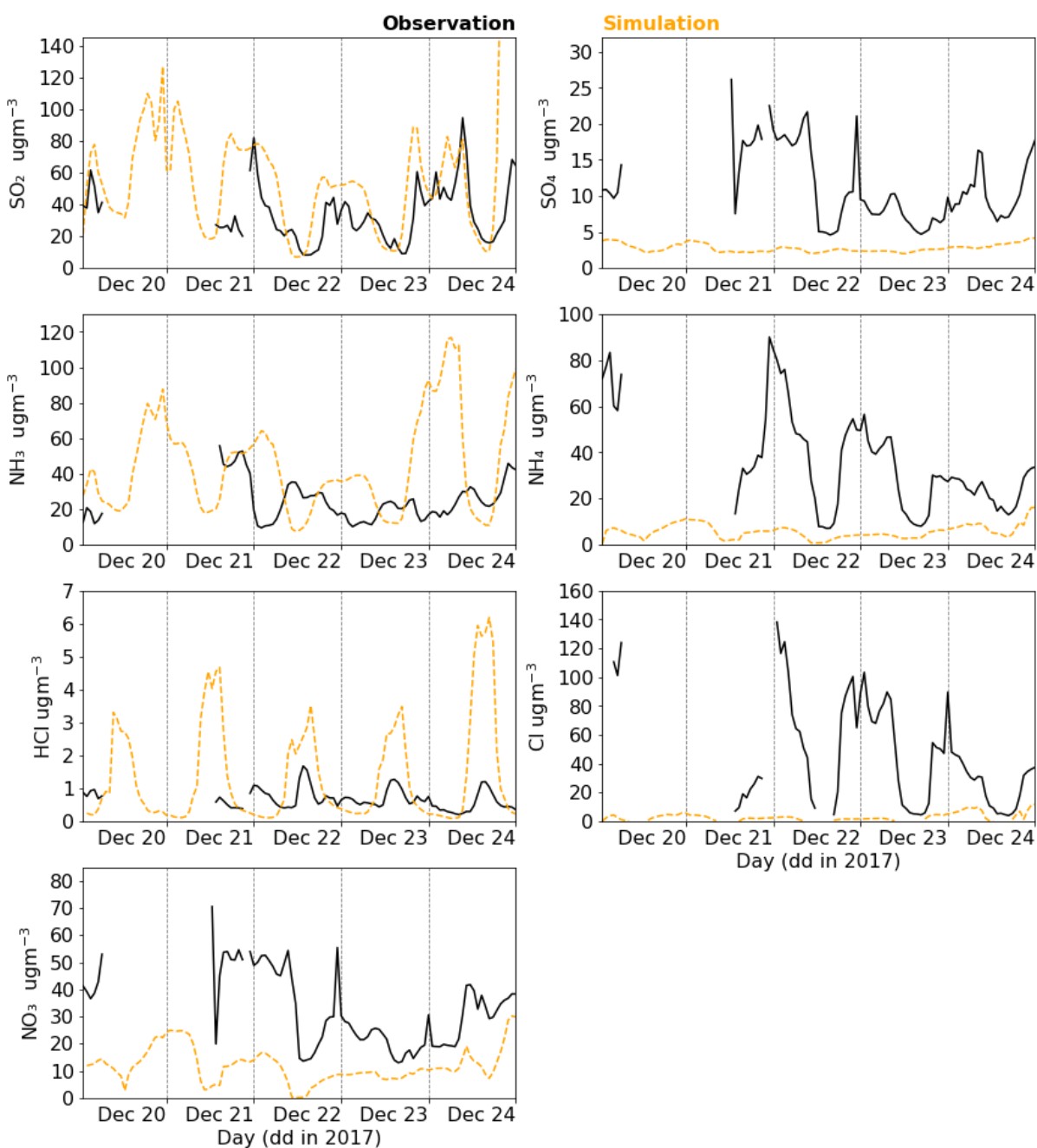

Figure 7 Comparison of WRF-Chem simulated ions ($SO_4^{2-}$, $NH_4^+$, $NO_3^-$, $Cl^-$) and trace gases ($SO_2$,
$NH_3$ & HCl) with the observation from WIFEX campaign at Delhi.



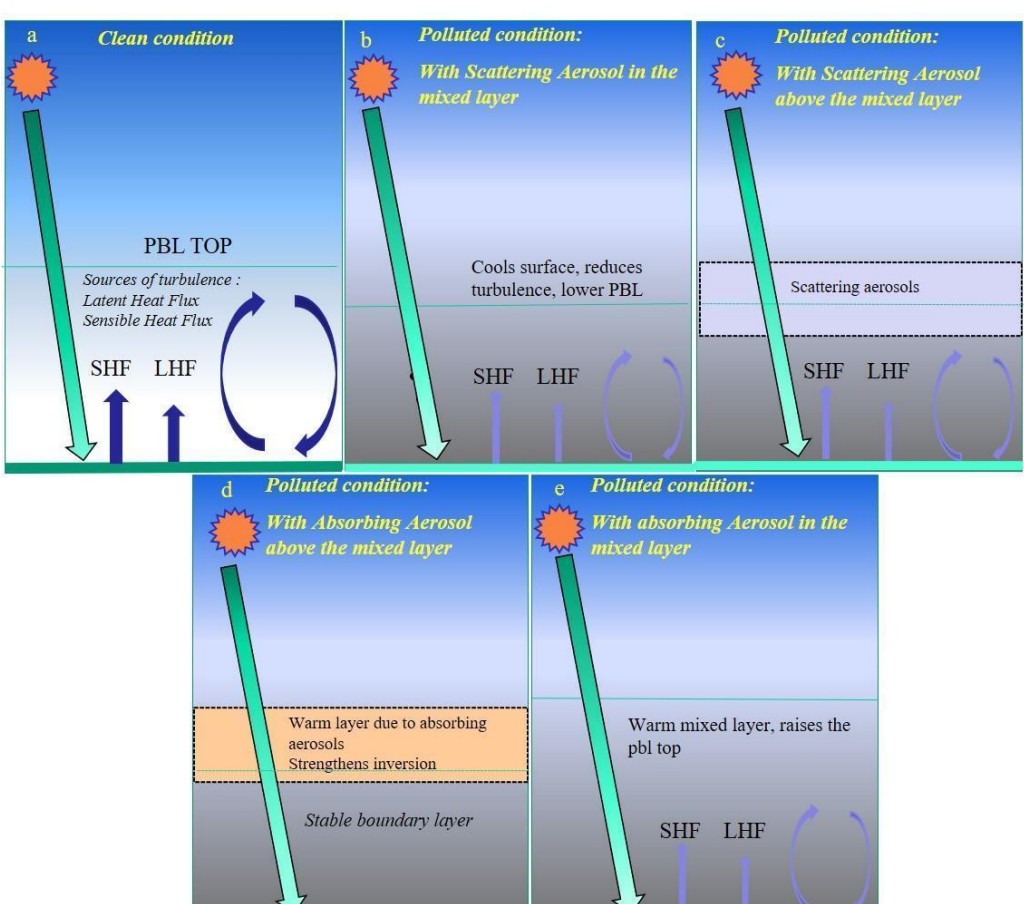


Figure 8 Schematic diagram of Aerosol Radiation Feedback.


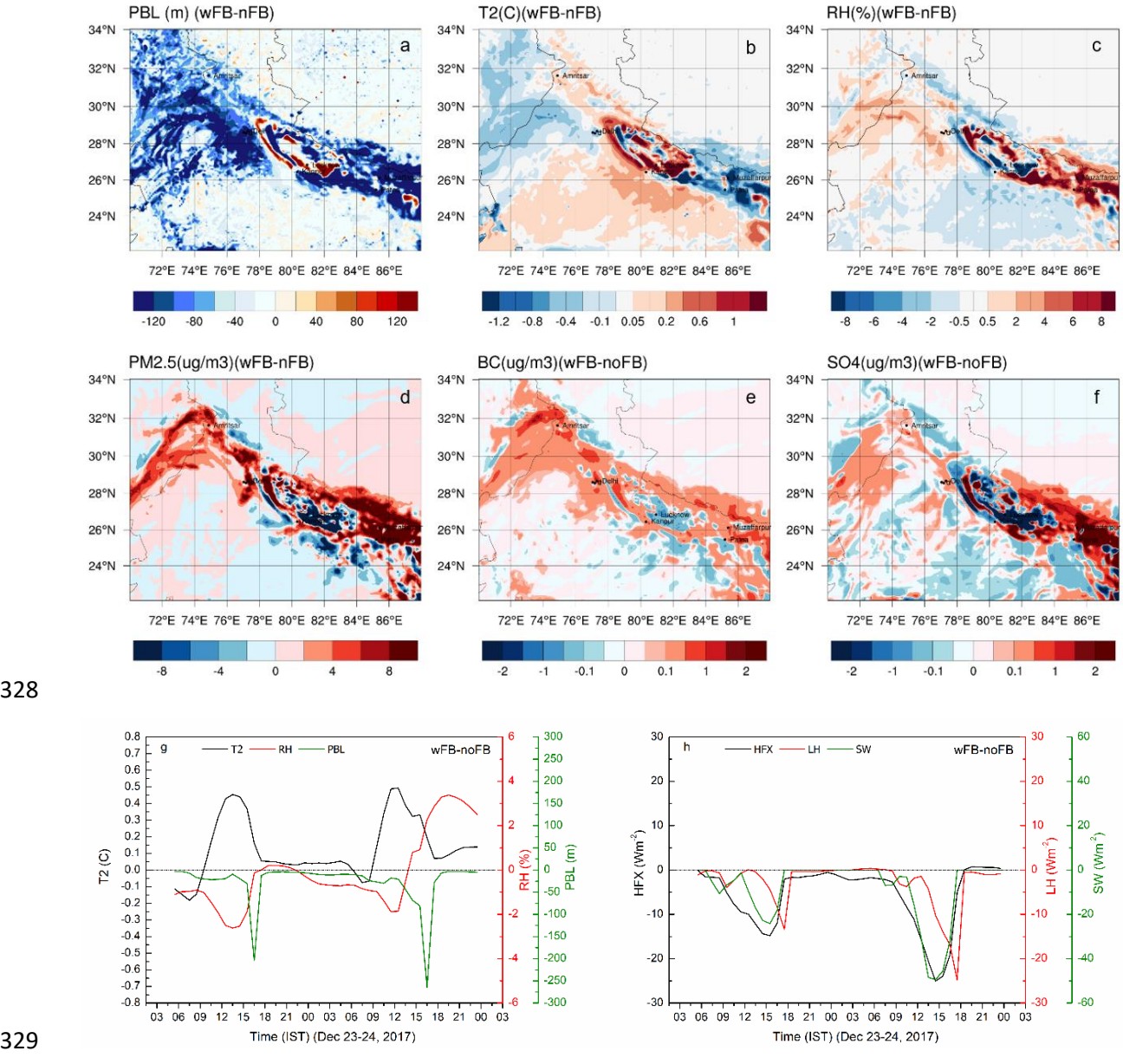



Figure 9 Effect of Aerosol Radiation feedback (wFB-nFB) on (a) PBL height, (b) 2-m temperature, (c) 2-m relative humidity, (d) surface PM$_{2.5}$, (e) surface BC and (f) surface SO$_4$ for December 24 at local noon (13:30-15:30 IST). (g) The time series of ΔPBL, ΔT2, and ΔRH; (h) ΔHFX (sensible heat flux), ΔLH (latent heat flux), and ΔSWF (downward shortwave flux) over CIGP for December 23 and 24. Δ denotes the difference between with and without AR feedback (wFB-nFB).







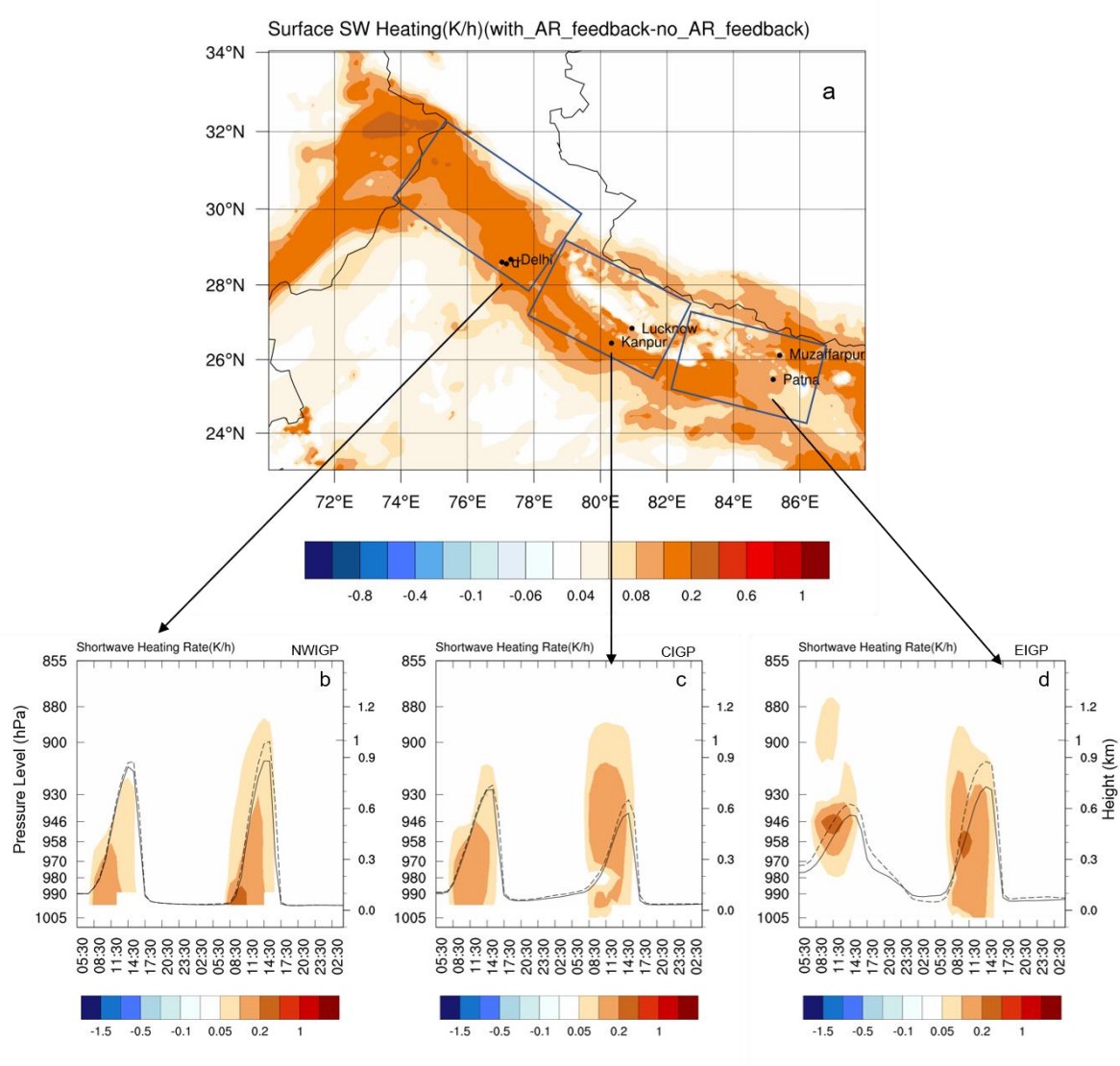


Figure 10 Differences in shortwave heating rates (K h[-1]) between simulations with and without aerosol radiation feedback (a) at the surface, and for pressure-time cross-sections over (b) NWIGP, (c) CIGP. And (d) EIGP for December 23 and 24. The solid and dashed lines are the PBL height with and without AR feedback respectively. The time is in IST.

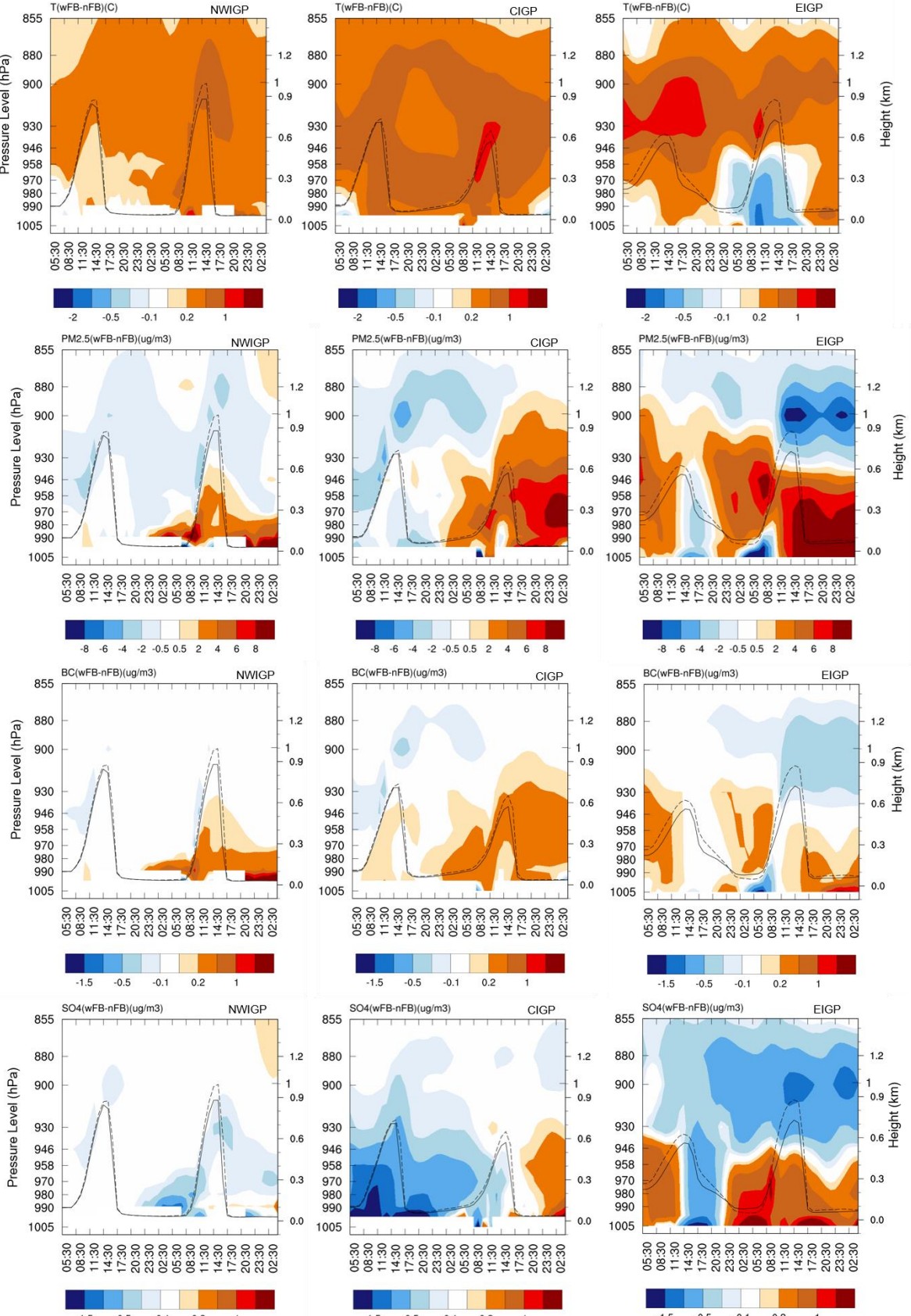




Figure 11 Pressure-time cross-section of the differences in T, $PM_{2.5}$, BC and $SO_4^{2-}$ between
simulations with and without the AR feedback for December 23 and 24. The solid and dashed lines
are the PBL height with and without AR feedback respectively. The time is in IST.


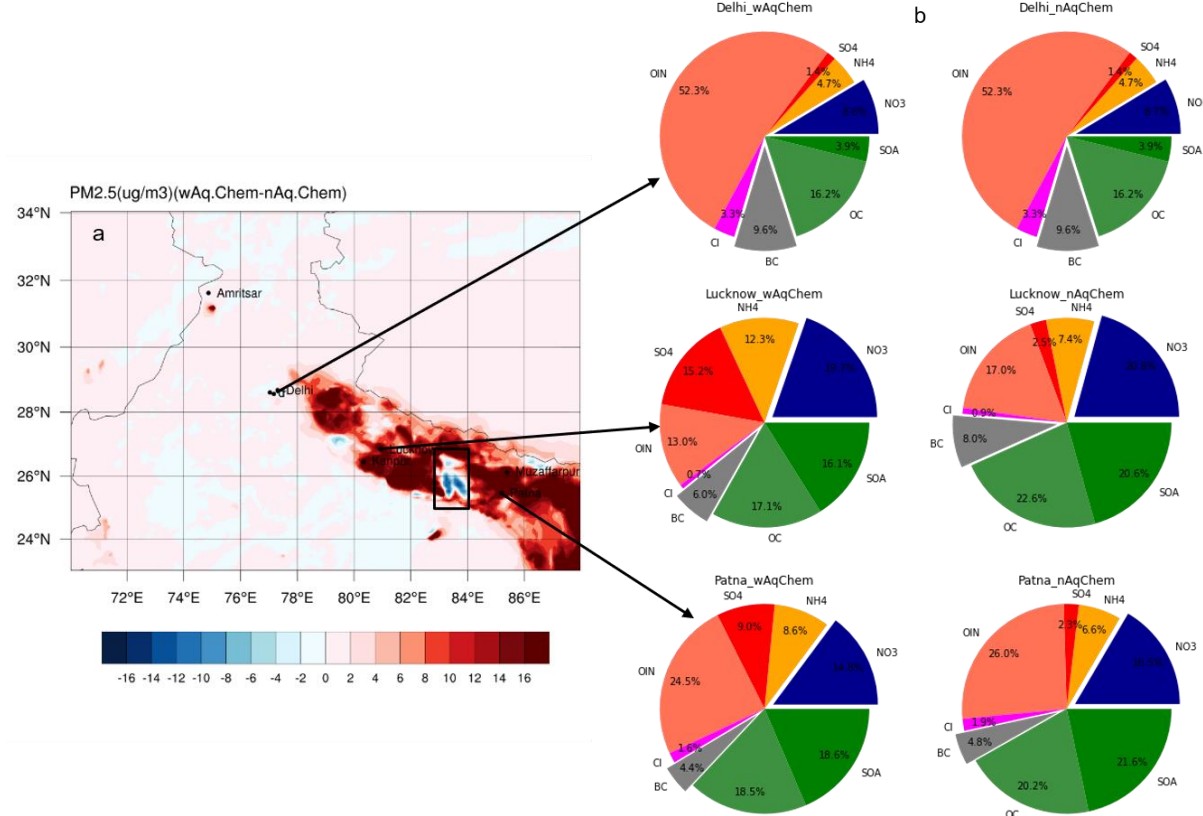


Figure 12 (a) Surface $\Delta PM_{2.5}$ (wAq.chem-noAq.chem) and (b) pie charts of $PM_{2.5}$ composition
distribution for the two cases, with and without Aqueous phase Chemistry for 24 Dec 2017. The
stations Delhi, Lucknow (LKN), and Patna are representative of NWIGP, CIGP, and EIGP regions
respectively.



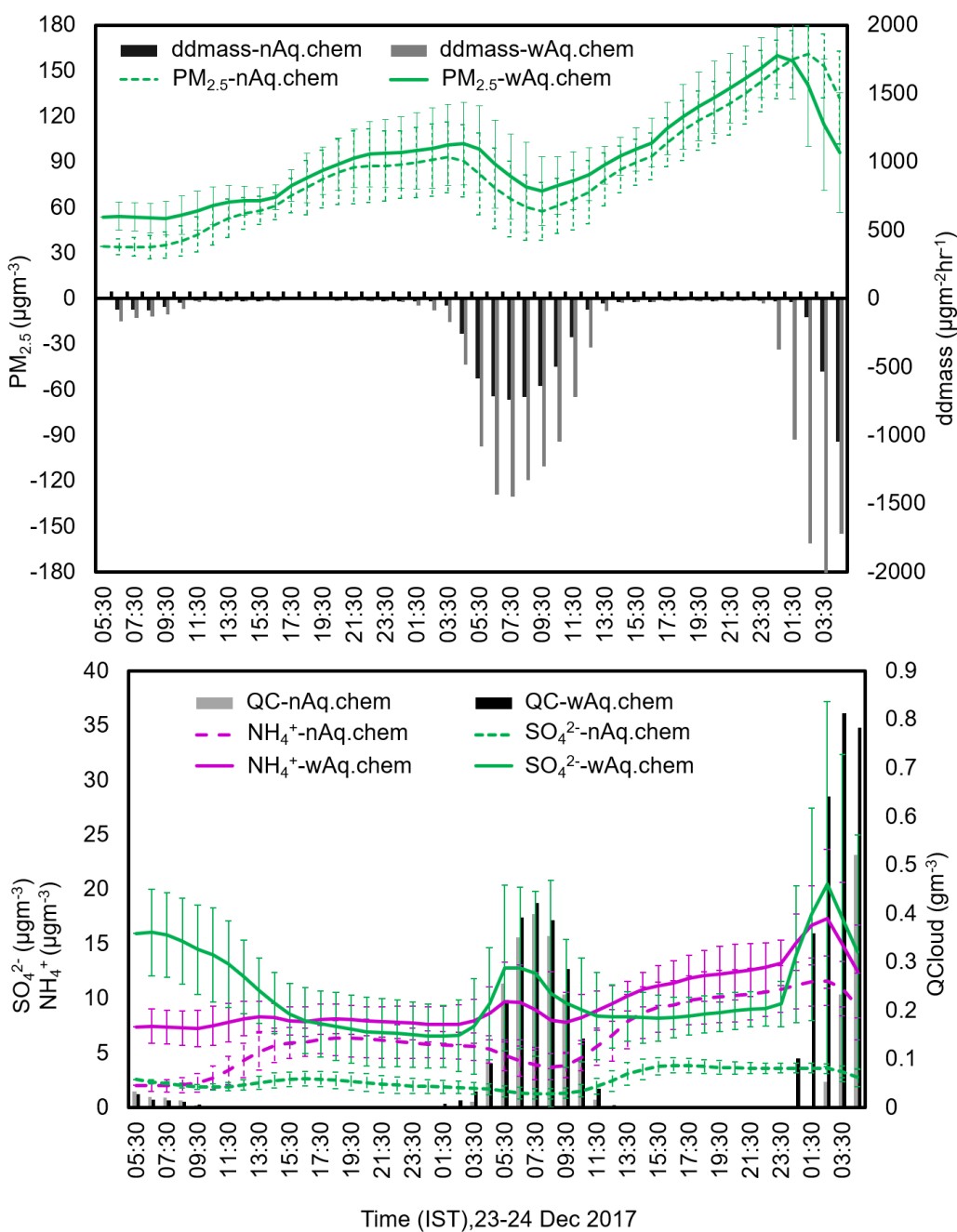


Figure 13 Time series of (a) $PM_{2.5}$ and its dry deposition (ddmass) flux change, (b) $SO_4^{2-}$, $NH_4^+$ and
LWC (QCloud) with and without aqueous phase chemistry included in the model, averaged over the
region bounded by a black rectangle in Fig. 12, for 23 and 24 December, 2017.


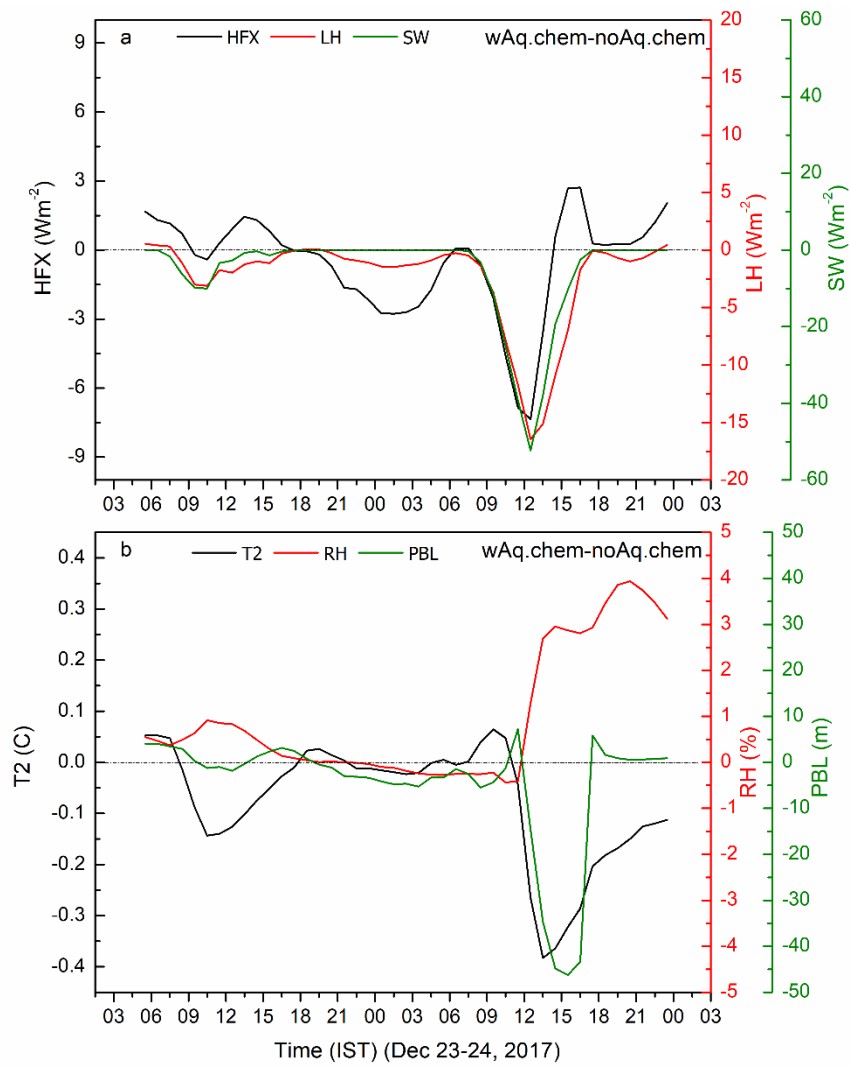



Figure 14 Time series of (a) ΔHFX (sensible heat flux), ΔLH (latent heat flux), and ΔSWF
(downward shortwave flux); (b) ΔT2, ΔRH, and ΔPBL over CIGP (79E-83E,26N-28N), for
23 and 24 December, 2017. Δ denotes the difference between with and without aqueous phase
chemistry.


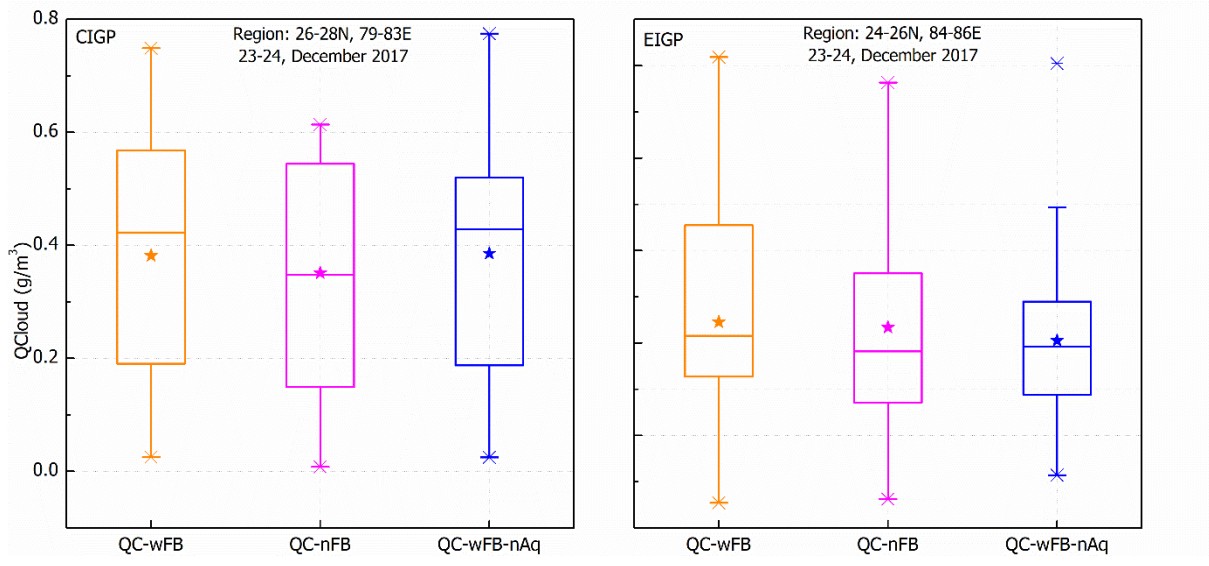



Figure 15 Averages (stars), medians (horizontal lines), quartiles (boxes), maxima, and minima for
LWC (QCloud) averaged over CIGP (left panel) and EIGP (right panel) for the fog event on 23-24
December 2017. Gold is for the simulation with AR feedback and aqueous chemistry, magenta for the
simulation with no AR feedback but includes aqueous chemistry, and blue for the simulation with AR
feedback but no aqueous chemistry. WRF-Chem does not produce fog in the NWIGP during the study
period.
















Table 2: Table showing the start and end time of fog1 on 23-24 December 2017 with LWC for the sensitivity experiments, with AR feedback, no AR feedback and no Aqueous phase chemistry

### Fog 1 (December 23-24, 2017)

| | EXP-wFB | | | EXP-nFB | | | EXP-nAq.Chem | | |
|---|---|---|---|---|---|---|---|---|---|
| | Start time (IST) | End time (IST) | Duration of Fog | Start time (IST) | End time (IST) | Duration of Fog | Start time (IST) | End time (IST) | Duration of Fog |
| **CIGP** | 16:30 | 15:30 | **23h** | 18:30 | 17:30 | **23h** | 18:30 | 15:30 | **21h** |
| LWC (g m$^{-3}$) | 0.034 | 0.036±0.032 | | 0.141±0.154 | 0.068±0.005 | | 0.184±0.138 | 0.034 ±0.021 | |
| Kanpur | 05:30 | 13:30 | **8h** | 05:30 | 12:30 | **7h** | 05:30 | 12:30 | **7h** |
| LWC (g m$^{-3}$) | 0.334± 0.487 | 0.017 | | 0.458±0.357 | 0.173± 0.071 | | 0.533 | 0.025± 0.0123 | |
| Lucknow | 23:30 | 14:30 | **15h** | 00:30 | 14:30 | **14h** | 23:30 | 14:30 | **15h** |
| LWC (g m$^{-3}$) | 0.269±0.145 | 0.087±0.040 | | 0.232±0.132 | 0.029±0.024 | | 0.139±0.084 | 0.025±0.012 | |
| **EIGP** | 21:30 | 12:30 | **15h** | 23:30 | 10:30 | **11h** | 21:30 | 10:30 | **13h** |
| LWC (g m$^{-3}$) | 0.099±0.092 | 0.007 | | 0.198±0.188 | 0.084±0.060 | | 0.026±0.008 | 0.153±0.119 | |
| Patna | 00:30 | 12:30 | **12h** | 04:30 | 10:30 | **6h** | 02:30 | 10:30 | **8h** |
| LWC (g m$^{-3}$) | 0.100±0.090 | 0.007 | | 0.009±0.005 | 0.038±0.041 | | 0.196±0.198 | 0.166±0.130 | |
| Muzzafarpur | 05:30 | 11:30 | **6h** | 06:30 | 10:30 | **4h** | 06:30 | 09:30 | **3h** |
| LWC (g m$^{-3}$) | 0.112±0.146 | 0.043±0.057 | | 0.051±0.041 | 0.003 | | 0.142±0.151 | 0.157±0.064 | |



Table 2: Table showing the start time of fog 2 on 24 December 2017 with LWC for the sensitivity
experiments, with AR feedback, no AR feedback and no Aqueous phase chemistry. Fog2 end time
could not be noted as simulation ended on 25 December 2017, 00UT (5:30 IST) before fog2
dissipates.

| Fog 2 (December 24, 2017) Start time (IST) | | |
| --- | --- | --- |
| **EXP-wFB** | **EXP-nFB** | **EXP-nAq.Chem** |
| **CIGP** 19:30 | **20:30** | **21:30** |
| LWC (g m$^{-3}$) 0.025 | 0.008±0.007 | 0.025 |
| Kanpur 21:30 | 22:30 | 23:30 |
| LWC (g m$^{-3}$) 0.041±0.007 | 0.298±0.218 | 0.482±0.398 |
| Lucknow 21:30 | 20:30 | 00:30 |
| LWC (g m$^{-3}$) 0.203±0.165 | 0.005 | 0.229±0.209 |
| **EIGP** 00:30 | **01:30** | **01:30** |
| LWC (g m$^{-3}$) 0.024±0.030 | 0.072±0.088 | 0.014±0.009 |
| Patna 03:30 | 03:30 | 03:30 |
| LWC (g m$^{-3}$) 0.030± 0.046 | 0.018 | 0.060 |
| Muzzafarpur 04:30 | No fog | No fog |
| LWC (g m$^{-3}$) 0.159±0.038 | | |


