# Peer review of "Role of atmospheric aerosols in severe winter fog over Indo Gangetic Plains of India: a"

_EGUsphere, 2023_

## Referee Comment (RC1)

This study uses the Weather Research and Forecasting model coupled with chemistry (WRF-Chem) to understand how (a) spatial distribution, mass, and composition of aerosols and (b) formation, duration and dissipation of fog is affected by aerosol-radiation feedback and aqueous-phase chemistry during a winter fog event over the highly polluted Indo-Gangetic Plain (IGP).

The paper is well written and well structured. The study is important and within the scope of the journal, but following major and minor comments need to be addressed before I can recommend the paper for publication.

Major Comments

1. The motivation to design EXP1, EXP2 and EXP3 is not very clear. Previous studies (for e.g., Mohan and Gupta, 2018) have already confirmed that ACM2 scheme performs better than other parameterizations. Does using a new PBL scheme improves the model performance or nudging the soil moisture improves the model performance?

2. L225: "Aerosol cloud interactions are not possible"- does that mean fog droplets do not activate? Explain. How realistic is fog lifecycle in EXP3 without 'Aerosol Cloud Interactions'?

3. Why Figure S1 only shows 6 stations? On that note, the authors need to explain how the gridded model results are compared to point observations? The wind speed observations are present only in 4 stations. Among that at Amritsar and at Delhi-RKP, wind speeds have large biases. The hypothesis of 'low measurement height and obstructions' needs some evidence. The authors should read previous literature and figure out how WRF-Chem performs in those regions? Are there any systematic biases?

4. The model performance in predicting mass and composition of PM2.5 is poor. The authors should add a table showing the correlation and error between observations and simulation and compare that with existing literature. Is it possible to design a sensitivity experiment by scaling up the rate of emissions (including the HCl emissions) in the EDGAR-HTAP inventory. Why inorganic ions (other than chloride) are also heavily underestimated and fails to predict the diurnal variations (sulfate for example)?

5. Both CIGP and EIGP regions are foggy. However, for the wFB simulation, in figure S5, the Single Scattering Albedo increases at EIGP but decreases at CIGP near the surface. Why?

6. When composition of different inorganic ions is discussed, is it an average over the entire sub-region? In that case a standard deviation value should also be reported. Are there previous similar studies from where the authors can

compare their results? Inside CIGP and EIGP, are there any differences in composition between the foggy and non-foggy grid boxes?

7. Why fog deposition is extremely important in the area marked by the box n Fig. 12a? and less important in other areas? Are the meteorological conditions different?

8. Why is the fog water content higher when aqueous chemistry is included?

9. Figure 15 needs to be modified and variations at three separate IGP regions need to be shown with a median and interquartile range, taking foggy grid boxes. Observations from the WiFex Campaign should also be added.

10. L559 mentions that Aerosol-Radiation feedback affects the timing of the fog. However, L584 mentions that aqueous phase chemistry together with radiation feedback promotes early fog formation. This is confusing. How does aqueous chemistry affect fog formation? Explain.

Minor Comments

1. The abstract needs to be shortened. The goal of the paper on L14 needs to be more specific. 'Aerosol-Radiation Interactions' and 'Aqueous Chemistry' should be mentioned in the beginning of the abstract.

2. L116: 'it is divided into three areas…'- Describe the areas. What are the spatial extends (latitude, longitude)? Which states/major cities included in each of the three areas?

3. L128: 'the WRF-Chem model version 4.0.3 has been used for this study'-cite a few studies who have used WRF-Chem or similar models to study aerosol-radiation feedback in fog.

4. L170: Avoid Italics. 'wFB-nFB' needs a bit more explaining. Write the expression in a way to make it more understandable to a general audience: for example: (Properties/ Parameters in wFB - Properties/ Parameters in nFB)

5. Figure1: True color MODIS reflectance map doesn't confirm presence of fog. Could they be low level clouds? Also, a different color (other than black) should be used to mark the stations.

6. The Taylor's diagram needs to be explained more carefully. How are the standardized normalized deviations calculated (include an equation)?

7. Add a table showing timings of the fog and liquid water content for all the sensitivity studies, for three different regions (with a mean and standard deviation). Also add mean and standard deviations of observations from all stations (for each area).

8. Figure 2: x and y-axis need proper labels. Also, can the authors use a color other than blue for EXP2?

9. What is 'ddmass'. Write clearly in figure captions.

10. 'ug/m3' should be changed to '$\mu g/m^3$'.

11. L522: Need citations for $PM_{2.5}$ compositions.
12. L567: change '01 UTC' to '01:00 UTC' and L 568: change '10 UTC' to '10:00 UTC'.

References

- Mohan, Manju & Gupta, Medhavi. (2018). Sensitivity of PBL parameterizations on PM 10 and ozone simulation using chemical transport model WRF-Chem over a sub-tropical urban airshed in India. Atmospheric Environment. 185. 10.1016/j.atmosenv.2018.04.054.

---

## Author Comment (AC1)

**This study uses the Weather Research and Forecasting model coupled with chemistry (WRF-Chem) to understand how (a) spatial distribution, mass, and composition of aerosols and (b) formation, duration and dissipation of fog is affected by aerosol-radiation feedback and aqueous-phase chemistry during a winter fog event over the highly polluted Indo-Gangetic Plain (IGP). The paper is well written and well structured. The study is important and within the scope of the journal, but following major and minor comments need to be addressed before I can recommend the paper for publication.**

We appreciate your thorough review and constructive comments on our manuscript. They have significantly contributed to the improvement and refinement of the work. Below, we provide a point by point response to your concerns. Your comments appear in the bold font and our responses are given in regular font.

**Major Comments**

**1. The motivation to design EXP1, EXP2 and EXP3 is not very clear. Previous studies (for e.g., Mohan and Gupta, 2018) have already confirmed that ACM2 scheme performs better than other parameterizations. Does using a new PBL scheme improves the model performance or nudging the soil moisture improves the model performance?**

Response:

The reviewer is correct in that Mohan and Gupta (2018) already showed that the ACM2 scheme performs better than other PBL parameterizations. However, they tested the YSU and ACM2 schemes during the summer time (1-15 June 2010) and focused on the evaluation of temperature, wind speed, PBL height, ozone, and $PM_{10}$. Since our period focuses on the winter season fog, it was critical for us to ensure that model captures all the relevant meteorological parameters reasonably well including relative humidity. That is why we designed EXP1, EXP2 and EXP3. Furthermore, to investigate both aerosol-radiation and aerosol-cloud interactions, the YSU (and other) PBL schemes in WRF-Chem can be used, but the ACM2 scheme is implemented in WRF-Chem differently than the other schemes making aerosol-cloud interactions not possible to investigate. That was our additional motivation to test the YSU PBL scheme. As shown in the manuscript, the YSU PBL scheme with NCEP initial and boundary conditions (IC/BC) underestimated the relative humidity preventing fog formation. Using the ERA-Interim IC/BC instead of NCEP improved the relative humidity predictions (Figure S1, supplement) but did not simulate fog well (Figure 3 of manuscript). Thus, following Mohan and Gupta (2018), the ACM2 PBL scheme was tested, giving predictions of fog comparable to observations (Figure 3 of manuscript). Including soil moisture nudging improved the surface fluxes and therefore the surface temperature and relative humidity. However, by switching to ACM2, only aerosol-radiation

interactions could be examined. Therefore, our results should be seen as complementing Mohan and Gupta (2018).

**2. L225: "Aerosol cloud interactions are not possible"- does that mean fog droplets do not activate? Explain. How realistic is the fog lifecycle in EXP3 without 'Aerosol Cloud Interactions'?**

Response:

In WRF-Chem, the aerosol-cloud interactions, that is, the dependence of cloud drop activation on the number of CCN which varies when using a prognostic aerosol scheme, is controlled by the mix-activate scheme that is called from a routine that computes vertical mixing of trace gases and aerosols. Because the ACM2 PBL scheme computes the vertical mixing of trace gases and aerosols and does not call mix-activate, it is not possible to examine the effect of aerosol-cloud interactions with our model configuration. Cloud (fog) water does form but is formed based on a prescribed CCN concentration that does not vary during the simulation.

Several modeling studies have confirmed that aerosol cloud interactions (ACI) plays important role in increasing fog liquid water content and lifetime, sometimes overweighing ARI in polluted environment (Maalick et al., 2016; Yan et al.,2021). Thus, any difference in the fog lifecycle due to ACI is not accounted for in our simulation.

3. **Why Figure S1 only shows 6 stations? On that note, the authors need to explain how the gridded model results are compared to point observations? The wind speed observations are present only in 4 stations. Among that at Amritsar and at Delhi-RKP, wind speeds have large biases. The hypothesis of 'low measurement height and obstructions' needs some evidence. The authors should read previous literature and figure out how WRF-Chem performs in those regions? Are there any systematic biases?**

Response:

We have compared the model output with observation (CPCB) for nine stations over IGP that are also considered for the statistical analysis in the Taylor diagram. However, to compare the time series of the meteorological parameters, only major cities as representative stations in each region of IGP have been shown. There were gaps in the wind speed dataset from CPCB for several stations of IGP, explaining why wind speed from only four stations are shown. At such low wind speeds, WRF-chem in general overestimates wind speed. The analysis code we use finds the WRF-Chem grid point nearest to the station's latitude and longitude. Several earlier studies have reported this bias in wind speed (Mohan and Gupta 2018; Pithani et al.,2019). However, this is unlikely to be the only or the dominant reason for the selective high bias as the photograph of the Amritsar monitoring station shows:

[Figure]

Figure 1: Meteorology monitoring station at Amritsar

For R. K Puram station, we have not been able to obtain a photograph of the station but it is located somewhere in the yellow polygon either along the main road (photo provided) or in the Sector 3&4 park, or in the campus of the National Institute of Health. All these locations have a lot of obstacles to free airflow. There is no large free space anywhere in that area. Moreover, WRF-Chem does not have the capability to represent building meteorology and parameterizes the effects of urban areas on meteorology through roughness length, which likely leads to overestimation of wind speed.

[Figure]

Figure 2: Map showing area around the meteorology monitoring station at RKPuram, Delhi.

**4. The model performance in predicting mass and composition of PM$_{2.5}$ is poor. The authors should add a table showing the correlation and error between observations and simulation and compare that with existing literature. Is it possible to design a sensitivity experiment by scaling up the rate of emissions (including the HCl emissions) in the EDGAR-HTAP inventory. Why inorganic ions (other than chloride) are also heavily underestimated and fails to predict the diurnal variations (sulfate for example)?**

Response:

Table. Statistics for $PM_{2.5}$ concentrations observed and modelled at four stations.

|  | Mean_PM$_{2.5}$ CPCB | Stdev | Mean_PM$_{2.5}$ WRFChem | Stdev | MB% NMB | RMSE | NMRSE | r | p |
|---|---|---|---|---|---|---|---|---|---|
| Amritsar | 83. 2 | 32.0 | 85.0 | 44.6 | -2.17 | 41.41 | 0.50 | 0.45 | 2.5E-07 |
| Delhi | 209 | 76.0 | 108 | 116. | 48.49 | 137 | 0.65 | 0.61 | 2.7E-13 |
| KNP | 232 | 141 | 109 | 74.8 | 53.02 | 172 | 0.74 | 0.52 | 1.3E-09 |
| Patna | 211 | 78.0 | 141 | 98.3 | 31.92 | 138 | 0.65 | 0.08 | 0.47 |

\* MB%= (Observed-Simulated)/Observed) \*100
\* NMRSE=RMSE/ Mean_PM$_{2.5}$ _CPCB

The statistical analysis between WRF-Chem results and observations (Table S1) shows a minimum mean bias for $PM_{2.5}$ at Amritsar (-2.2%) while at other stations it ranges from 48 to 53% similar to the reported range of model bias (underestimated by 40–60%) in winter over IGP by earlier studies (Bran and Srivastava, 2017;Ojha et al., 2020). The RMSE values range from 41 to 138 µg/m$^3$ (normalized RMSE~0.4 to 0.7) are comparable to the reported values by these studies. The Pearson correlation coefficient (r) for the simulated and observed day-to-day variation in $PM_{2.5}$ lies between 0.4 and 0.7 for all the stations in Fig. 6 except at Patna which lies within the range in these studies. The poor correlation at Patna is due to the loss of $PM_{2.5}$ during fog in the model as discussed earlier.

This is one of the few studies where aerosol composition has been evaluated for the IGP. WRF-Chem fails to predict chemical composition correctly, especially the chloride concentration. It is hypothesized that the aerosol chloride is from trash burning, thus we incorporated trash-burning emissions which include HCl emissions from  Chaudhury et al.,(2021). However, the inventory only covers trash burning that occurs due to lack of waste management infrastructure, and does not include the well documented use of trash as wintertime heating fuel (Nagpure et al., 2015). Adding trash burning emissions increased $PM_{2.5}$ concentrations (comparison not shown in manuscript), however, a large bias between simulated and observed $PM_{2.5}$ still exists. The underestimation is likely caused by a low bias in the residential burning emission inventory and a failure of the emission inventory to represent residential sector emissions (including HCl emissions) from both cow dung burning and from the use of trash as cheap heating fuel properly. Since solid fuel is the second largest source for several species including $SO_2$ and $NH_3$ this affects the model measurement agreement of all species. Punjab, being a relatively prosperous state with a comparatively high per capita LPG refilling rate shows the lowest discrepancies between the measurement and the modelled data. In Uttar Pradesh and Bihar, the reliance on cheap solid fuels is highest among all Indian states, and discrepancies are larger. In Delhi missing industrial sources may also add to the model measurement discrepancies. Since the present residential emission inventory lacks residential sector HCl emissions and fails to reproduce current spatial emission patterns, a scaling is

unlikely to solve the problem. Emission inventories still propagate the patterns prevalent in the times before LPG usage became an indicator of social status. Consequently, they have the highest per capita residential sector emissions in prosperous states such as Punjab (where people used to be able to afford 3 meals a day) and lower per capita emissions in poorer states (where cooking used to be restricted to once per day). This spatial pattern has now reversed, but no fundamental inventory emission inventory revision has happened. Instead, old spatial patterns are just scaled to keep pace with population growth.

A study by Pawar et al., (2023 ) showed an improvement in ammonium prediction by scaling HCl (3xHCl) demonstrating the uncertainties in the chemistry scheme of the model and that model improvements are required to correctly simulate PM composition. To better understand why WRF-Chem poorly predicts aerosol composition, extensive work needs to be pursued by obtaining more observations and conducting other sensitivity simulations like the one suggested by the reviewer. This work (and others e.g. Pawar et al., 2023) is simply pointing out the need for further work to understand the processes affecting aerosol composition in the IGP.

5. **Both CIGP and EIGP regions are foggy. However, for the wFB simulation, in figure S5, the Single Scattering Albedo increases at EIGP but decreases at CIGP near the surface. Why?**

Response: Figures 9 and 11 in the manuscript show the differences in scattering and absorbing aerosols (i.e., sulfate and BC). In EIGP, sulfate concentrations have a larger concentration with AR feedback than without AR feedback with time periods where the difference is >1 ug/m$^3$ (Fig.11). The BC concentration changes are small (<0.5 µg/m3) in the EIGP, resulting in a higher SSA near the surface with AR feedback in EIGP. In the CIGP, BC concentrations increase while sulfate aerosols decrease within the PBL with AR feedback (Fig.11) compared to the simulation without AR feedback. A decrease in SSA is seen for the CIGP throughout the boundary layer while in EIGP the decrease occurs near the top of the PBL. Also contributing to the higher SSA in EIGP is the increase in RH (Fig. 9) due to AR feedback which favours the growth of aerosols in size by uptake of water and the production of secondary aerosols such as $SO_4^{2-}$ and $NH_4^+$.

6. **When composition of different inorganic ions is discussed, is it an average over the entire sub-region? In that case a standard deviation value should also be reported. Are there previous similar studies from where the authors can compare their results? Inside CIGP and EIGP, are there any differences in composition between the foggy and non-foggy grid boxes?**

Response: Timeseries of inorganic ions in Fig 13 is an average over the region bounded by the box in Fig 12a. The standard deviation has been added to the figure as suggested.

Gupta and Mandariya.,(2013) showed a decrease in PM$_1$ in fog due to wet scavenging by fog droplets (referred in the discussion on timeseries of PM$_{2.5}$). The box plot in Fig.3 below shows the difference in the inorganic composition ($SO_4^{2-}$ and $NH_4^+$) with and without aqueous phase chemistry in foggy and

non-foggy grids of CIGP and EIGP. The inorganic ions concentration is higher with aqueous phase chemistry than without aqueous phase chemistry in both the foggy and non-foggy grids in both the regions. The difference is larger in foggy grids compared to the non-foggy grids for both $SO_4^{2-}$ and $NH_4^+$. However, $SO_4^{2-}$ shows large variability in foggy grid points and with aqueous phase chemistry, likely due to both the formation and loss processes occurring in fog as shown in Fig 13 and discussed in section 6 of manuscript. This indicates that $SO_4^{2-}$ is largely formed in fog through aqueous phase chemistry.

[Figure]

Figure 3: Box plot showing $SO_4^{2-}$ and $NH_4^+$ with and without aqueous phase chemistry included in the model, averaged for the region CIGP and EIGP for foggy and non-foggy grids.

7. **Why fog deposition is extremely important in the area marked by the box in Fig. 12a? and less important in other areas? Are the meteorological conditions different?**

Response: The region bounded by the box has dense fog because of increased RH with the AR feedback with a maximum difference of ~ 4.5% compared to the surrounding region (<2%). As discussed in the manuscript, the inclusion of aqueous phase chemistry produces and adds sulfate to the aerosol mass, which will increase the $PM_{2.5}$ fog deposition flux. and increase the role of $PM_{2.5}$ on radiation for the simulation with AR feedback. Increase in $PM_{2.5}$ contributes to AR feedback and, consequently, fog intensity increases. Also note that deposition of aerosol with cloud water increases in dense fog.

8. **Why is the fog water content higher when aqueous chemistry is included?**

Response: It is discussed in section 6.

We observe an increase in $PM_{2.5}$ (esp. $SO_4^{2-}$ and $NH_4^+$) when aqueous phase chemistry is included in the model. The aqueous chemistry increases both the mass of $PM_{2.5}$ and the size of the aerosols, both of which contribute to AR feedback, thus increasing RH and PBL stability. The increase in RH also saturates the air, promotes aerosol growth by water uptake, and thus favors fog formation.

9. **Figure 15 needs to be modified and variations at three separate IGP regions need to be shown with a median and interquartile range, taking foggy grid boxes. Observations from the WiFex Campaign should also be added.**

Response: A box plot is shown below for WRF-Chem simulated QCloud (QC) taking foggy grid boxes with AR feedback, no AR Feedback, and no Aqueous phase chemistry experiments over CIGP and EIGP. We do not have fog over NWIGP in the model for the study period. Each box plot represents the median (50th percentile), with the box indicating the interquartile range (IQR) between the 25th and 75th percentiles. QC is comparatively higher with AR feedback than without AR feedback and without aqueous phase chemistry for both CIGP and EIGP. IQR is larger for simulation with and without AR feedback than without aqueous phase chemistry in CIGP showing large variability in the LWC. Whereas in EIGP the variability in LWC is greater in simulation with AR feedback compared to the other two experiments. We also observe an increase in fog LWC with AR feedback and aqueous phase chemistry compared to without AR feedback and the absence of aqueous phase chemistry (Fig.5).

[Figure]

Figure 4: Box plot showing LWC at the surface for the three experiments, with and without AR feedback and without aqueous phase chemistry for CIGP and EIGP. NWIGP does not have fog during the study period.

WiFEx has observation for aerosols and fog at Delhi, while the WRF-Chem simulation does not have fog in NWIGP. Therefore, the WiFEx observations are not included in the plot.

[Figure]

Figure5. Map showing LWC at the surface for the three experiments, with and without AR feedback, without aqueous phase chemistry and their anomalies.

**10. L559 mentions that Aerosol-Radiation feedback affects the timing of the fog. However, L584 mentions that aqueous phase chemistry together with radiation feedback promotes early fog formation. This is confusing. How does aqueous chemistry affect fog formation? Explain.**

Response: The aqueous chemistry increases both the mass of $PM_{2.5}$ and the size of the aerosols, both of which contribute to AR feedback. Consequently, the AR feedback affects the PBL dynamics affecting the T2 and RH near the surface. This further affects the fog formation and dissipation. We noted in Table 1 and 2 (in response to minor comments 7) that the timing of fog with AR feedback, without AR feedback, and without aqueous phase chemistry are different with formation occurring earlier in the simulation with AR feedback which also includes aqueous phase chemistry. It is likely due to the contribution of added $PM_{2.5}$ to AR feedback when aqueous phase chemistry is included.

**Minor Comments**

**1. The abstract needs to be shortened. The goal of the paper on L14 needs to be more specific. 'Aerosol-Radiation Interactions' and 'Aqueous Chemistry' should be mentioned in the beginning of the abstract.**

Response: Abstract have been modified in the manuscript.as suggested.

**2. L116: 'it is divided into three areas…'- Describe the areas. What are the spatial extends (latitude, longitude)? Which states/major cities included in each of the three areas?**

Response: The three regions are defined by the following latitude-longitude range which has also been added in the text.

'it is divided into three areas, northwest (NWIGP: latitude-longitude range, 27°N-32°N,75°E-79°N), central (CIGP: latitude-longitude range, 25°N-28°N,79°E-83°E), and east (EIGP: latitude-longitude range, 24°N-27°N, 83°E-87°E)'. Discussion on major cities have been added to the Observations section as suggested by Reviewer 2.

**3. L128: 'the WRF-Chem model version 4.0.3 has been used for this study'-cite a few studies who have used WRF-Chem or similar models to study aerosol radiation feedback in fog.**

Response: WRF-Chem has been widely used to study the role of aerosol radiation (AR) on air quality and haze episodes. References have been added in the text as follows:

"Earlier studies have successfully used WRFChem to predict fog (Pithani et al., 2019) and in the study of aerosol-radiation feedback on air quality (Kumar et al., 2020; Bharali et al., 2019) and the study of aerosol-fog interactions (Shao et al., 2023)."

**4. L170: Avoid Italics. 'wFB-nFB' needs a bit more explaining. Write the expression in a way to make it more understandable to a general audience: for example: (Properties/ Parameters in wFB - Properties/ Parameters in nFB)**

Response: Modified as suggested:

Impact of radiation feedback=Parameters in wFB- Parameters in nFB

Impact of aqueous phase chemistry= Parameters in wAq.chem- Parameters in noAq.chem

**5. Figure1: True color MODIS reflectance map doesn't confirm presence of fog. Could they be low level clouds? Also, a different color (other than black) should be used to mark the stations.**

Response: True color MODIS reflectance map represents low cloud indicative of likely fog. These cloud layers align with the fog imagery from INSAT 3D; therefore, we used the image to represent fog. Color of text marking stations changed from black to yellow.

**6. The Taylor's diagram needs to be explained more carefully. How are the standardized normalized deviations calculated (include an equation)?**

Response: The following equation to calculate standardized deviations in Taylor diagram has been added:

The centered RMS difference, the correlation, and the standard deviation are related by the following formula:

$E'^2 = \sigma_o^2 + \sigma_m^2 - 2\sigma_o\sigma_m R$

where R is the correlation coefficient between the model-simulated and observed fields, E' is the centered RMS difference between the fields, and $\sigma_m^2$ and $\sigma_o^2$ are the variances of the model-simulated and observed fields, respectively. The correlation(R), centered RMS difference (E') and standard deviations of the model simulated and observed fields are calculated by the following formulas:

$$R = \frac{\frac{1}{N}\sum(M_n - \overline{M})(O_n - \overline{O})}{\sigma_m\sigma_o} \qquad (1)$$

$$E'^2 = \frac{1}{N}\sum[(M_n - \overline{M}) - (O_n - \overline{O})]^2 \qquad (2)$$

$$\sigma_m^2 = \frac{1}{N}\sum(M_n - \overline{M})^2 \qquad (3)$$

$$\sigma_o^2 = \frac{1}{N}\sum(O_n - \overline{O})^2 \qquad (4)$$

where the overall mean of a field is indicated by an overbar.

**7. Add a table showing timings of the fog and liquid water content for all the sensitivity studies, for three different regions (with a mean and standard deviation). Also, add mean and standard deviations of observations from all stations (for each area).**

Response: A table has been added in the manuscript listing the start and end time of two fog events with LWC for the sensitivity experiments, with AR feedback, no AR feedback and no Aqueous phase chemistry. The table includes fog for the two regions CIGP and EIGP along with two stations in each region. NWIGP does not have fog during the study period. The table along with the box plot of fog in response to major comment 9 above shows that fog intensity and duration changes with AR feedback compared to that without AR feedback and without aqueous phase chemistry for both Fog1 and Fog2. Fog forms earlier with AR feedback compared to the simulation without AR feedback in CIGP as well as EIGP, while it dissipates earlier with AR feedback in CIGP and delayed in EIGP compared to that without AR feedback and without aqueous phase chemistry. In both the regions, the fog lifetime increases with AR feedback. All the stations, however do not show the same pattern, for example, fog 1 in Luckow forms and dissipates at the same time for simulations with AR feedback and without aqueous phase chemistry, and fog2 forms later with AR feedback than without AR feedback. Patna shows no difference in fog2 formation in all the three experiments. We recommend that simulations at higher spatial and temporal resolutions would be better to study the fog dynamics at point locations.

Table 1: Table showing the start and end time of fog1 on 23-24 December 2017 with LWC for the sensitivity experiments, with AR feedback, no AR feedback and no Aqueous phase chemistry

**Fog 1 (December 23-24, 2017)**

| | EXP-wFB | | | EXP-nFB | | | EXP-nAq.Chem | | |
|---|---|---|---|---|---|---|---|---|---|
| | Start time (IST) | End time (IST) | Duration of Fog | Start time (IST) | End time (IST) | Duration of Fog | Start time (IST) | End time (IST) | Duration of Fog |
| **CIGP** | 16:30 | 15:30 | **23h** | 18:30 | 17:30 | **23h** | 18:30 | 15:30 | **21h** |
| LWC (g/m$^3$) | 0.034 | 0.036±0.032 | | 0.141±0.154 | 0.068±0.005 | | 0.184±0.138 | 0.034 ±0.021 | |
| Kanpur | 05:30 | 13:30 | **8h** | 05:30 | 12:30 | **7h** | 05:30 | 12:30 | **7h** |
| LWC (g/m$^3$) | 0.334± 0.487 | 0.017 | | 0.458±0.357 | 0.173± 0.071 | | 0.533 | 0.025± 0.0123 | |
| Lucknow | 23:30 | 14:30 | **15h** | 00:30 | 14:30 | **14h** | 23:30 | 14:30 | **15h** |
| LWC (g/m$^3$) | 0.269±0.145 | 0.087±0.040 | | 0.232±0.132 | 0.029±0.024 | | 0.139±0.084 | 0.025±0.012 | |
| **EIGP** | 21:30 | 12:30 | **15h** | 23:30 | 10:30 | **11h** | 21:30 | 10:30 | **13h** |
| LWC (g/m$^3$) | 0.099±0.092 | 0.007 | | 0.198±0.188 | 0.084±0.060 | | 0.026±0.008 | 0.153±0.119 | |
| Patna | 00:30 | 12:30 | **12h** | 04:30 | 10:30 | **6h** | 02:30 | 10:30 | **8h** |
| LWC (g/m$^3$) | 0.100±0.090 | 0.007 | | 0.009±0.005 | 0.038±0.041 | | 0.196±0.198 | 0.166±0.130 | |
| Muzzafarpur | 05:30 | 11:30 | **6h** | 06:30 | 10:30 | **4h** | 06:30 | 09:30 | **3h** |
| LWC (g/m$^3$) | 0.112±0.146 | 0.043±0.057 | | 0.051±0.041 | 0.003 | | 0.142±0.151 | 0.157±0.064 | |

Table 2: Table showing the start time of fog 2 on 23-24 December 2017 with LWC for the sensitivity experiments, with AR feedback, no AR feedback and no Aqueous phase chemistry

| | Fog 2 (December 24, 2017) Start time (IST) | | |
|---|---|---|---|
| | **EXP-wFB** | **EXP-nFB** | **EXP-nAq.Chem** |
| **CIGP** | **19:30** | **20:30** | **21:30** |
| LWC (g/m$^3$) | 0.025 | 0.008±0.007 | 0.025 |
| Kanpur | 21:30 | 22:30 | 23:30 |
| LWC (g/m$^3$) | 0.041±0.007 | 0.298±0.218 | 0.482±0.398 |
| Lucknow | 21:30 | 20:30 | 00:30 |
| LWC (g/m$^3$) | 0.203±0.165 | 0.005 | 0.229±0.209 |
| **EIGP** | **00:30** | **01:30** | **01:30** |
| LWC (g/m$^3$) | 0.024±0.030 | 0.072±0.088 | 0.014±0.009 |
| Patna | 03:30 | 03:30 | 03:30 |
| LWC (g/m$^3$) | 0.030± 0.046 | 0.018 | 0.060 |
| Muzzafarpur | 04:30 | No fog | No fog |
| LWC (g/m$^3$) | 0.159±0.038 | | |

• Fog still exists when the simulation ended on 25 December 2017, 00UT (5:30 IST)

**8. Figure 2: x and y-axis need proper labels. Also, can the authors use a color other than blue for EXP2?**

Response: Figure 2 has been modified as suggested

**9. What is 'ddmass'. Write clearly in figure captions.**

Response: 'ddmass' is dry deposition flux

**10. 'ug/m3' should be changed to 'μg/m3'.**

Response: Corrected in the MS

**11. L522: Need citations for PM2.5 compositions.**

Response: Citations have been added in the MS

**12. L567: change '01 UTC' to '01:00 UTC' and L 568: change '10 UTC' to '10:00 UTC'.**

Response: Time format has been changed in the MS

---

## Author Comment (AC2)

**The authors simulated a fog event with high aerosol loading over the Indo Gangetic Plains of India, and estimated the role of atmospheric aerosols in severe winter fog through aerosol-radiation interaction and aqueous phase chemistry. Overall, this is well-executed study and the topic is very interesting. The authors made great efforts in model evaluation and sensitivity simulation. However, several conclusions in the main text should be further explained and the presentation quality can be improved. I suggest a major revision before it can be accepted.**

We appreciate your thorough review and constructive comments on our manuscript. They have significantly contributed to the improvement and refinement of the work. Below, we provide a point by point response to your concerns. Your comments appear in the bold font and our responses are given in regular font.

**Major comments:**

**I am very curious about if the fog event is due to radiation or advection. If this is a radiation fog, then the role of AR feedback could be a major reason. However, if this is an advection fog, the authors may need to pay attention on wind changes.**

Response: It is a radiation fog. The majority of fog events in the IGP during December-January are radiation fog (Deshpande et al., 2023; Ghude et al.,2023) formed due to radiative cooling of the surface.

**Simulated PM2.5. In L363, the authors mentioned that the observed high PM2.5 can be predicted by model but not for PM2.5 composition (Fig.7). I am very confused about this. The model strongly underestimated observed inorganic PM2.5 composition on 24th Dec; then why was total PM2.5 mass concentration well simulated?**

Response: The WRF-Chem model results show that a large percentage of $PM_{2.5}$ is classified as "other inorganics (OIN)", which is usually dominated by $PM_{2.5}$ other than BC and OC. OIN contributes to >50% of $PM_{2.5}$ in Delhi ( Fig 12 of manuscript). Emissions of mineral dust from road and construction dust also contribute to aerosol loading in urban areas (Sharma and Mandal.,2023). The model simulates the day-to-day variability in $PM_{2.5}$ considerably well, still, there is a large bias between the measured and observed values due to discrepancy in the simulation of $PM_{2.5}$ composition and other factors discussed in the MS and quantified in the new Table S1.

**Decreased PM2.5 in CIGP due to AR feedback. The authors mentioned this issue in L427-436 but didn't explain the reason clearly. How about the changes in winds due to AR, which may be a reason**.

Response: Over the CIGP, the AR feedback causes a depletion of surface $PM_{2.5}$, which is likely due to their hygroscopic growth, and then dry deposition (e.g. average dry deposition mass concentration of $PM_{2.5}$=331 µg/m²/hr with AR feedback and 282 µg/m²/hr without AR feedback on 24th December) in

dense fog. The increase in RH with AR feedback favors the growth of aerosols in size by the uptake of water.

Wind speed increases and $PM_{2.5}$ decreases with AR feedback on December 23, whereas wind speed decreases and $PM_{2.5}$ increases on December 24 (Fig.1 below) which is expected due to reduced dispersion in the stable boundary layer. However, there is a reduction in $PM_{2.5}$ (>10 µg/m³) on 24 December between 00:30 and 03:30 hrs, which is likely due to loss via deposition in fog and not due to change in wind speed with AR feedback.

[Figure]

Figure 1. Time series showing the difference in $PM_{2.5}$ and windspeed due to AR feedback over CIGP (79-83E,26-28N).

**High particulate Cl. The model can't reproduce the observed high Cl that could come from trash-burning and industrial emissions. I am wondering if trash-burning emissions from Chaudhary et al. (2021) can represent Cl emissions at a city scale.**

Response: The reviewer makes a good point in that the trash-burning emissions inventory may not represent trash burning in urban regions that need high horizontal grid spacing to capture variations among neighbourhoods in the city. The trash-burning emissions of Chaudhary et al., (2021) are available at 10 km resolution and capture city-scale emissions of larger megacities like Delhi, which occupy several grid cells. However, the inventory contains annual emissions and fails to resolve the seasonality of trash-burning emissions identified by Nagpure et al., (2015). They suggested almost all the waste-burning emissions in neighbourhoods with higher socioeconomic status in Delhi occur due to the use of waste as cheap heating fuel by individuals such as night watchmen and pavement dwellers. Hence these "rich" neighbourhoods see little to no waste burning for most of the year (≤18 kg/km²-day), except during winter (~89 kg/km²-day). Hence wintertime waste-burning emissions, in central Delhi, may be underestimated in the Chaudhary et al., (2021) inventory, which only considers waste burning that occurs due to lack of collection infrastructure, and at landfills and, therefore, shows a concentration of waste burning emissions around the periphery of the Delhi NCR, but low waste burning emissions in the relatively prosperous City centre. During radiation fog events with very limited transport, such a

discrepancy in the local emissions can indeed drive a model measurement discrepancy. However, such "heating motivated" waste-burning emissions would ideally need to be included in an updated residential sector emission inventory, which should also capture the seasonality of residential sector emissions caused by heating demand and hot water needs.

In addition, emissions from other sources (e.g., industries) are not accounted for in WRF-Chem and the WRF-Chem chemistry option used does not represent halogen chemistry. The results shown in this paper suggest that further work on HCl emissions needs to be conducted in order to improve aerosol composition in the IGP region.

**Fog duration. In Fig.15, I find the results from three simulations are quite similar and I suggest the authors to clearly identify their difference**.

Response: To better quantify the differences among the three simulations, a box plot and table have been added comparing the three simulations as also suggested by reviewer 1, which show that the average fog intensity and duration of fog increases comparatively with AR feedback than without AR feedback and without aqueous phase chemistry. Discussions added in the MS

**Minor comments:**

**L18-20: the model tends to strongly underestimate observed PM2.5.**

Response: The line has been modified as suggested.

**L33: "These processes" refers to aerosol-radiation interaction and aqueous phase chemistry? The latter can't change PBL meteorology.**

Response: Aqueous phase chemistry cannot directly change the PBL meteorology, however the increased $PM_{2.5}$ concentrations and size of the aerosols caused by the aqueous phase chemistry affect the PBL dynamics through AR Feedback and hence fog formation

**L42: "NOx" refers to emission or concentration?**

Response: NOx here refers to $NO_2$ column concentration. It has been rephrased accordingly in the text.

**L47&L55: duplication for NAAQS**

Response: Duplicate has been removed.

**L123-125: I suggest to move this description to Section 2.2 Observations**

Response: Moved to Section 2.2 Observations as suggested

**L183: please rephrase "aromatic compounds HONO"**

Response: rephrased as "aromatic compounds, HONO"

**L270-272: again, this information should be given in the Observations part if not.**

Response: Removed, as it is already mentioned in Section 2.2 Observations

**L300: in Fig.2, I find most of the correlation coefficient (r) for RH is below 0.87. so I am not convinced by the "r2>0.75".**

Response: It is the correlation coefficient **r** and not **r²**. Thank you for bringing it to our attention. "r2>0.75" corrected as "r>0.75".

**L308: please change "are" to "is".**

Response: Corrected in the MS as suggested

**L351-352: I suggest the authors to show the correlation coefficient using daily data to justify this argument.**

Response: A table has been added to show the statistics between simulated and measured $PM_{2.5}$.

**Section4: It looks the authors failed to discuss about the uncertainties in chemistry scheme used in the model, which is also an importance source of model biases.**

Response:  Discussion added in the MS.

**L532: In fact, the model can't reproduce the observed aerosol composition. The following discussions on previous studies couldn't be helpful.**

Response: We agree with the reviewer that the model could not reproduce the observed aerosol chemical composition. We took several steps to improve the model performance as described below. First, we tested three different meteorological configurations as discussed in the manuscript and used the best configuration for conducting sensitivity simulations focused on exploring aerosol-fog interactions. Second, we included the best estimates of trash-burning emissions to represent anthropogenic chloride aerosols in our configuration. While incorporation of trash burning emissions did improve the model simulations of $PM_{2.5}$ and better captured the day-to-day variability of $PM_{2.5}$, we still see large underestimation of chloride aerosols in the model. This indicates the need for more work on better quantifying trash burning emissions, which may not only improve particulate chloride in the model but also improve simulations of other aerosol chemical components through aerosol thermodynamics. Despite these challenges, the model was able to simulate the same changes in inorganic composition during fog events as reported by observational studies referred to in the MS. This encouraged us to perform sensitivity simulations described in the manuscript. We anticipate that the challenges identified in our study will provide motivation to both our and other groups working on this part of the world to close this gap between observations and model simulations.

---

## Referee Report (RR1)

Thank you for responding to my comments. The responses have significantly improved the manuscript. Based on the responses, I have additional comments listed below.

- Supplementary Table S1: Please show the statistics for all the 9 stations separately. Normalized Mean Bias percentage is defined as:
  $$NMB\ (\%) = \frac{\sum(Model - Observation)}{\sum Observation} \times 100\%.$$
  Please update the equation and recalculate the numbers. Kindly change 'MB %' to 'NMB %' in the header. Update the main text accordingly. Why is the p value for Patna so high compared to other locations? Kindly include the number of datapoints used to calculate the statistics. Change 'NMRSE' to 'NRMSE'.

- A sensitivity study was conducted by including trash burning emissions. Please add a line commenting on the NMB value for $PM_{2.5}$ concentrations in that study.

- Including the equations for the Taylor diagram is helpful. I further suggest the authors to pick any station: for example, pick station 2 for EXP2 and guide the readers about what specific information can be obtained from Figure 3.

- The equation for centered RMS is repeated twice. Please fix. What is the significance of the centered RMS?

- The authors claim that a PBL scheme which works fine for summer over IGP, might not work for winter, as a motivation for designing EXP1, 2, and 3: Are there any known seasonal biases in the PBL schemes used for this study? A short description is needed. The authors might also add a few lines (or a table) briefly describing the differences between different PBL schemes used in this study? (See Xie et al., 2012, for example. They recommended using ACM2 PBL scheme for both summer and winter, for a different, but highly polluted region)

- A few lines should be added in the conclusion/discussion section about the inability of the model to simulate the aerosol-fog interactions and the potential affects it might have on the outcome on the paper. On that note, why the exchange co-efficient of heat is needed to calculate the activation fraction? Kindly refer to relevant equations/literature for better clarity.

- How do the authors identify whether a fog event is Radiation Fog or an Advection Fog (Both from the observations and the model)?

- L397: Fix grammar. Change Mean Bias to Normalized Mean Bias.

- Figure 1c: I think the units are not required for the title. Please change the title to: "Anthropogenic $PM_{2.5}$ Emissions". Please increase the gap between texts: 'Kanpur' and 'Lucknow'. Change kg/m$^2$/s to kg m$^{-2}$ s$^{-1}$.

- In Line 349, and in the caption of Figure 3, please clarify if the cloud water mixing ratios are grid average or in-cloud (i.e. divided by cloud fraction). Kindly incorporate the same change throughout the manuscript.
- L667: change "diagnostic output.." to "diagnostic output.
- L445: Typically, what percentage of $PM_{2.5}$ mass is secondary in the IGP? How much is nitrate?
- L487: I recommend changing µg/m2/hr to $\mu g\ m^{-2}\ hr^{-1}$. How accurate is the dry deposition flux in the model? Cite previous work, if available.
- L603: Fix grammar.
- L672: Please explain "more CCN are expected with aqueous chemistry"
- Kindly change ug/m3 to $\mu g\ m^{-3}$ or $\mu g/m^3$ throughout the manuscript.
- WRF-Chem simulations does not have fog in NWIGP, and hence could not be compared with the WiFEx campaign data: Please add a few lines in the conclusion/discussion section.
- L12: Improve the sentence structure.
- How are the representative stations selected? Are there data available only from the 9 stations across the IGP during the study period?

References:

Xie, B., J. C. H. Fung, A. Chan, and A. Lau (2012), Evaluation of nonlocal and local planetary boundary layer schemes in the WRF model, *J. Geophys. Res.*, 117, D12103, doi:10.1029/2011JD017080.

---

## Author Response (AR2)

*We appreciate the reviewer's thorough review and constructive comments on our manuscript. Below, we provide a point-by-point response to your concerns. Your comments appear in the regular font and our responses are given in italics font.*

Thank you for responding to my comments. The responses have significantly improved the manuscript. Based on the responses, I have additional comments listed below.

1. Supplementary Table S1: Please show the statistics for all the 9 stations separately. Normalized Mean Bias percentage is defined as:

   a. $$NMB\,(\%) = \frac{\sum(Model-Observation)}{\sum Observation} \times 100\%.$$

   b. Please update the equation and recalculate the numbers. Kindly change 'MB %' to 'NMB %' in the header. Update the main text accordingly. Why is the p value for Patna so high compared to other locations? Kindly include the number of datapoints used to calculate the statistics. Change 'NMRSE' to 'NRMSE'.

   *Response: We missed the summation symbol in the equation. Thank you for bringing this to our attention. The table has now been updated.*

   *The following discussions have been added to section 4 of the manuscript*

   *The poor correlation at Patna (and Muzaffarpur) is due to the low modeled $PM_{2.5}$ concentrations which are caused by increased dry deposition of aerosol particles activated as fog droplets during fog periods, as discussed in section 4 of the manuscript. Furthermore, the fog events in WRF and observations have somewhat different time periods causing WRF-predicted $PM_{2.5}$ and the observed $PM_{2.5}$ concentrations to decrease at different times.*

2. A sensitivity study was conducted by including trash burning emissions. Please add a line commenting on the NMB value for $PM_{2.5}$ concentrations in that study.

   *Response: The following text has been added in section 4 of the manuscript.*
   *This was accomplished by incorporating trash-burning emissions in the model simulation, which improved the $PM_{2.5}$ prediction, increasing NMB by ~4-8% in IGP*

   | Stations | NMB%-with Trash emissions | NMB%-no Trash emissions | Difference in NMB% |
   |---|---|---|---|
   | *Amritsar* | *2.17* | *-2.52* | *4.69* |
   | *Dwarka (Delhi)* | *-48.49* | *-52.71* | *4.22* |
   | *IHBAS(Delhi)* | 31.93 | 24.71 | *7.22* |
   | *RKP(Delhi)* | -40.44 | -45.7 | *5.22* |
   | *Kanpur* | *-53.01* | *-57.65* | *4.63* |
   | *Lucknow* | -30.14 | -32.49 | *2.35* |
   | *Patna* | *-32.31* | *-40.69* | *8.37* |
   | *Muzaffarpur* | -36.46 | -40.45 | *4* |

3. Including the equations for the Taylor diagram is helpful. I further suggest the authors to pick any station: for example, pick station 2 for EXP2 and guide the readers about what specific information can be obtained from Figure 3.

   *Response: Discussion added as suggested in the manuscript.*

   *"For example, simulated RH at Dwarka (4) and Lucknow (7) for EXP2, and IGI Airport (2), IHBAS (3), Lucknow (7), and Patna (8) for EXP3 show good agreement with observation, with r>0.7, standard deviation within ±0.25 and mean bias within 10%. Among these stations, the model performs better for Dwarka (4) and Lucknow (7) for EXP2, IGI Airport (2), and IHBAS (3) for EXP3 with a smaller centered*

*RMSE (<0.75). "*

*"For example, simulated T2 agrees best with observation at IHBAS (3) for EXP1 and IGI Airport (2) for EXP2, with smaller centered RMSE and standard deviation, and bias <5%."*

4. The equation for centered RMS is repeated twice. Please fix. What is the significance of the centered RMS?

   *Response: Revised in the manuscript.*

   *The centered RMS difference is between the modelled and observed datasets proportional to the distance of a point in the Taylor diagram to the point "OBS" on the x-axis, indicating the extent to which the simulated datasets compare with the observed dataset. It is calculated by centering both the datasets around their respective means.*

5. The authors claim that a PBL scheme which works fine for summer over IGP, might not work for winter, as a motivation for designing EXP1, 2, and 3: Are there any known seasonal biases in the PBL schemes used for this study? A short description is needed. The authors might also add a few lines (or a table) briefly describing the differences between different PBL schemes used in this study? (See Xie et al., 2012, for example. They recommended using ACM2 PBL scheme for both summer and winter, for a different, but highly polluted region)

   **Response:** *The following paragraph has been added to the manuscript.*
   *Although earlier studies (Gunwani and Mohan, 2017; Mohan and Bhati, 2011; Mohan and Gupta, 2018; Xie et al., 2012) recommend using the nonlocal ACM2 PBL scheme for air quality prediction for IGP, there is still seasonal, day-night and regional biases in the PBL schemes. Gunwani and Mohan, (2017) showed that ACM2, QNSE, and MYJ schemes work well in predicting temperature, humidity, and wind speed in different regions. ACM2, MYNN and MYJ work best for Chennai (in South India), New Delhi (NWIGP), and Kolkata (EIGP) respectively for PBL height during summer whereas for winter MYJ works best for Chennai and QNSE for New Delhi and Kolkata. Regarding the prediction of fog, Mohan and Bhati, (2011) found that using ACM2 PBL scheme with Pleim Xiu surface physics improved wintertime meteorology estimates in Delhi indicating its potential in fog predictions, whereas Pithani et al.,(2019) recommend using the local PBL scheme MYNN2.5 with WSM3, WSM6, and Lin microphysics. Shin and Hong, (2011) found that a non-local (e.g., ACM2, YSU) scheme is favorable in unstable conditions and a local scheme (e.g., MYJ, Boulac) in stable conditions. All these studies suggest the need for careful consideration of the above-mentioned biases while selecting a PBL scheme.*

   *The YSU and ACM2 PBL schemes are both nonlocal schemes, however, studies report differences in their performance particularly in the convective daytime boundary layer, with a deeper boundary layer height using ACM2 compared to YSU (Hariprasad et al., 2014; Xie et al., 2012). This is likely due to their different formulations e.g. defining the critical bulk Richardson number (Xie et al., 2012).*

6. A few lines should be added in the conclusion/discussion section about the inability of the model to simulate the aerosol-fog interactions and the potential affects it might have on the outcome on the paper. On that note, why the exchange co-efficient of heat is needed to calculate the activation fraction? Kindly refer to relevant equations/literature for better clarity.

   *Response:*
   *The following text was added to the conclusions.*
   *Aerosol-cloud interactions were not investigated in this study due to the limitation of the ACM2 PBL scheme in providing necessary information with other modules in WRF. Previous studies of aerosol-fog interactions have found that ACI also promotes early onset of fog formation and increases fog duration (Maalick et al., 2016; Yan et al., 2021). While these previous studies were applied to midlatitude fog events, it is likely that ACI also plays a dominant role in IGP fogs, suggesting that future studies are needed to fully understand aerosol effects on IGP fog events.*

*The mixactivate module in WRF-Chem computes the activation fraction for aerosol mass and number based on when the maximum supersaturation of the air entering the cloud exceeds the critical supersaturation to form cloud droplets based on Kohler theory (Abdul-Razzak and Ghan, 2000, 2002). The maximum supersaturation relies on the mean vertical velocity and the turbulent velocity spectrum as input. The turbulent velocity spectrum depends on the heat exchange coefficient, which causes the spectrum of vertical velocities. Thus, the cloud droplet activation scheme relies on information from the PBL scheme. Unfortunately, the ACM2 PBL scheme does not provide the heat exchange coefficient to other parts of the WRF code, so aerosol-cloud interaction using ACM2 is not possible.*

*We added text to the model description to note why the exchange coefficient of heat is needed.*

7. How do the authors identify whether a fog event is Radiation Fog or an Advection Fog (Both from the observations and the model)?

   *Response: Radiation fog is formed when the surface cools and humidity levels reach 100%, particularly at night under the clear sky and calm winds (Lakra and Avishek, 2022). Radiation fog is usually categorized based on the onset time of fog, which occurs after sunset and before sunrise. Advection fog on the other hand occurs when horizontally warm, moist air moves over cooler surfaces. Earlier studies categorized advection fog based on visibility and wind speed, where reduced visibility was accompanied by wind speeds exceeding 2.5 m/s, followed by a sudden visibility decrease, indicating an advection-type fog event (Deshpande et al., 2023; Pithani et al., 2019). The fog event in our study is a wintertime fog that starts to form at ~20:00 LT. The nighttime wind speeds were <2.5 m/s at the stations shown in Figure S1, further supporting that the fog events studied were radiation fog. The majority of fog events in the IGP during December-January are radiation fog (Deshpande et al., 2023; Ghude et al., 2023) formed due to radiative cooling of the surface, with longer-duration events compared to other regions of the world (Deshpande et al., 2023).*
   *Lines 120-121 rephrased in section 2. Methodology of the manuscript*

8. L397: Fix grammar. Change Mean Bias to Normalized Mean Bias.

   *Response: Sentence corrected in the manuscript.*

9. Figure 1c: I think the units are not required for the title. Please change the title to: "Anthropogenic PM$_{2.5}$ Emissions". Please increase the gap between texts: 'Kanpur' and 'Lucknow'. Change kg/m$^2$/s to kg m$^{-2}$ s$^{-1}$.

   *Response: Suggested changes are implemented in the Figure 1 in the manuscript.*

10. In Line 349, and in the caption of Figure 3, please clarify if the cloud water mixing ratios are grid average or in-cloud (i.e. divided by cloud fraction). Kindly incorporate the same change throughout the manuscript.

    *Response: Cloud water mixing ratios are grid average and it is defined in the manuscript text and Figure 3.*

11. L667: change "diagnostic output.." to "diagnostic output.
    *Response: Change implemented in the manuscript.*

12. L445: Typically, what percentage of PM$_{2.5}$ mass is secondary in the IGP? How much is nitrate?
    *Response: PM$_{2.5}$ composition varies across the Indo-Gangetic Plain (IGP). For example, Sharma and Mandal, (2017) reported that secondary aerosols contribute to 23% of PM$_{2.5}$ mass in Delhi, whereas Behera and Sharma, (2010) found that 50% of PM$_{2.5}$ is secondary aerosols, 34% of secondary inorganic aerosol (SIA) and 17% of secondary organic aerosol (SOA) in Kanpur. Another study estimated that the*

*total secondary aerosols contribute to 42 ± 10% of PM$_{2.5}$ in winter and 23 ± 6% in summer (Nagar et al., 2017) Nitrate constituted 9-13% of PM$_{2.5}$ mass in Delhi (Lalchandani et al., 2021; Sharma and Mandal, 2017).*

13. L487: I recommend changing µg/m2/hr to µg m$^{-2}$ hr$^{-1}$. How accurate is the dry deposition flux in the model? Cite previous work, if available.

   *Response:*

   *Units revised in the manuscript*

   *We do not have observations to validate dry deposition flux. Dry deposition of gases and aerosols is a process that needs continued evaluation and is a focus of some past and future studies. In WRF-Chem, the dry deposition of gas species is calculated following Wesely, (1989) while aerosol dry deposition follows Binkowski and Shankar, (1995). Ryu and Min, (2022) found higher dry deposition velocity for coarse mode particles in the model, resulting in the underestimation of surface PM$_{10}$ concentration. The updated dry deposition scheme by Ryu et al., (2022) significantly increased surface PM$_{10}$ concentrations but showed minimal impact on PM$_{2.5}$ levels. Although this study was done for another region, it is reasonable to assume that the dry deposition flux of PM$_{2.5}$ over IGP is within acceptable limits. In addition, the AQMEII project is currently conducting an evaluation of dry and wet deposition (Galmarini et al., 2021). This model intercomparison study will provide valuable information on deposition model parameterizations.*

14. L603: Fix grammar.
   *Response: Sentence corrected in the manuscript.*

15. L672: Please explain "more CCN are expected with aqueous chemistry"

   *Response: The aqueous chemistry adds sulfate to the aerosol mass increasing the mass of PM$_{2.5}$. Increased PM$_{2.5}$ further contributes to AR feedback, thus increasing RH. Increased RH favors the growth of aerosol size which then promotes the availability of aerosols as CCN.*

   *We have explained the phrase in section 7 of the manuscript*

16. Kindly change ug/m3 to µg m$^{-3}$ or µg/m$^3$ throughout the manuscript.
   *Response: Unite revised throughout the manuscript.*

17. WRF-Chem simulations does not have fog in NWIGP, and hence could not be compared with the WiFEx campaign data: Please add a few lines in the conclusion/discussion section.

   *Response: Suggested lines added in the manuscript.*

18. L12: Improve the sentence structure.
   *Response: Sentence corrected in the manuscript*

19. How are the representative stations selected? Are there data available only from the 9 stations across the IGP during the study period?

   *Response: Representative stations were selected based on the availability of data. CPCB (Central Pollution Control Board of India) has a large network of stations throughout the country, particularly Delhi in the IGP. However, most of the stations in other parts of IGP experienced gaps in data during the winter of 2017. At present, CPCB data availability has improved, and it can be verified at the CPCB website*

*(https://airquality.cpcb.gov.in/ccr/#/caaqm-dashboard-all/caaqm-landing/caaqm-data-availability)*

*Text rephrased in section 2.2 Observations of the manuscript*

References:

Abdul-Razzak, H. and Ghan, S. J.: A parameterization of aerosol activation: 2. Multiple aerosol types, J. Geophys. Res. Atmos., 105(D5), 6837–6844, doi:https://doi.org/10.1029/1999JD901161, 2000.

Abdul-Razzak, H. and Ghan, S. J.: A parameterization of aerosol activation 3. Sectional representation, J. Geophys. Res. Atmos., 107(D3), AAC 1-1-AAC 1-6, doi:https://doi.org/10.1029/2001JD000483, 2002.

Behera, S. N. and Sharma, M.: Reconstructing primary and secondary components of PM2.5 composition for an Urban Atmosphere, Aerosol Sci. Technol., 44(11), 983–992, doi:10.1080/02786826.2010.504245, 2010.

Binkowski, F. S. and Shankar, U.: The Regional Particulate Matter Model: 1. Model description and preliminary results, J. Geophys. Res. Atmos., 100(D12), 26191–26209, doi:https://doi.org/10.1029/95JD02093, 1995.

Deshpande, P., Meena, D., Tripathi, S., Bhattacharya, A. and Verma, M. K.: Event-based fog climatology and typology for cities in Indo-Gangetic plains, Urban Clim., 51, 101642, doi:https://doi.org/10.1016/j.uclim.2023.101642, 2023.

Galmarini, S., Makar, P., Clifton, O. E., Hogrefe, C., Bash, J. O., Bellasio, R., Bianconi, R., Bieser, J., Butler, T., Ducker, J., Flemming, J., Hodzic, A., Holmes, C. D., Kioutsioukis, I., Kranenburg, R., Lupascu, A., Perez-Camanyo, J. L., Pleim, J., Ryu, Y. H., San Jose, R., Schwede, D., Silva, S. and Wolke, R.: Technical note: AQMEII4 Activity 1: Evaluation of wet and dry deposition schemes as an integral part of regional-scale air quality models, Atmos. Chem. Phys., 21(20), 15663–15697, doi:10.5194/acp-21-15663-2021, 2021.

Ghude, S. D., Jenamani, R. K., Kulkarni, R., Wagh, S., Dhangar, N. G., Parde, A. N., Acharja, P., Lonkar, P., Govardhan, G., Yadav, P., Vispute, A., Debnath, S., Lal, D. M., Bisht, D. S., Jena, C., Pawar, P. V., Dhankhar, S. S., Sinha, V., Chate, D. M., Safai, P. D., Nigam, N., Konwar, M., Hazra, A., Dharmaraj, T., Gopalkrishnan, V., Padmakumari, B., Gultepe, I., Biswas, M., Karipot, A. K., Prabhakaran, T., Nanjundiah, R. S. and Rajeevan, M.: WiFEX Walk into the Warm Fog over Indo-Gangetic Plain Region, Bull. Am. Meteorol. Soc., 104(5), E980–E1005, doi:10.1175/BAMS-D-21-0197.1, 2023.

Gunwani, P. and Mohan, M.: Sensitivity of WRF model estimates to various PBL parameterizations in different climatic zones over India, Atmos. Res., 194(2016), 43–65, doi:10.1016/j.atmosres.2017.04.026, 2017.

Hariprasad, K. B. R. R., Srinivas, C. V., Singh, A. B., Vijaya Bhaskara Rao, S., Baskaran, R. and Venkatraman, B.: Numerical simulation and intercomparison of boundary layer structure with different PBL schemes in WRF using experimental observations at a tropical site, Atmos. Res., 145–146, 27–44, doi:10.1016/j.atmosres.2014.03.023, 2014.

Lakra, K. and Avishek, K.: A review on factors influencing fog formation, classification, forecasting, detection and impacts, Springer International Publishing., 2022.

Lalchandani, V., Kumar, V., Tobler, A., M. Thamban, N., Mishra, S., Slowik, J. G., Bhattu, D., Rai, P., Satish, R., Ganguly, D., Tiwari, S., Rastogi, N., Tiwari, S., Močnik, G., Prévôt, A. S. H. and Tripathi, S. N.: Real-time characterization and source apportionment of fine particulate matter in the Delhi megacity area during late winter, Sci. Total Environ., 770, doi:10.1016/j.scitotenv.2021.145324, 2021.

Maalick, Z., Kühn, T., Korhonen, H., Kokkola, H., Laaksonen, A. and Romakkaniemi, S.: Effect of aerosol concentration and absorbing aerosol on the radiation fog life cycle, Atmos. Environ., 133, 26–33, doi:10.1016/j.atmosenv.2016.03.018, 2016.

Mohan, M. and Bhati, S.: Analysis of WRF Model Performance over Subtropical Region of Delhi, India, Adv. Meteorol., 2011, 1–13, doi:10.1155/2011/621235, 2011.

Mohan, M. and Gupta, M.: Sensitivity of PBL parameterizations on PM10 and ozone simulation using chemical transport model WRF-Chem over a sub-tropical urban airshed in India, Atmos. Environ., 185, 53–63, doi:10.1016/j.atmosenv.2018.04.054, 2018.

Nagar, P. K., Singh, D., Sharma, M., Kumar, A., Aneja, V. P., George, M. P., Agarwal, N. and Shukla, S. P.: Characterization of PM2.5 in Delhi: role and impact of secondary aerosol, burning of biomass, and municipal solid waste and crustal matter, Environ. Sci. Pollut. Res., 24(32), 25179–25189, doi:10.1007/s11356-017-0171-3, 2017.

Pithani, P., Ghude, S. D., Prabhakaran, T., Karipot, A., Hazra, A., Kulkarni, R., Chowdhuri, S., Resmi, E. A., Konwar, M., Murugavel, P., Safai, P. D., Chate, D. M., Tiwari, Y., Jenamani, R. K. and Rajeevan, M.: WRF model sensitivity to choice of PBL and microphysics parameterization for an advection fog event at Barkachha, rural site in the Indo-Gangetic basin, India, Theor. Appl. Climatol., 136(3–4), 1099–1113, doi:10.1007/s00704-018-2530-5,

2019.

Ryu, Y. H. and Min, S. K.: Improving Wet and Dry Deposition of Aerosols in WRF-Chem: Updates to Below-Cloud Scavenging and Coarse-Particle Dry Deposition, J. Adv. Model. Earth Syst., 14(4), doi:10.1029/2021MS002792, 2022.

Sharma, S. K. and Mandal, T. K.: Chemical composition of fine mode particulate matter (PM2.5) in an urban area of Delhi, India and its source apportionment, Urban Clim., 21, 106–122, doi:10.1016/j.uclim.2017.05.009, 2017.

Shin, H. H. and Hong, S. Y.: Intercomparison of Planetary Boundary-Layer Parametrizations in the WRF Model for a Single Day from CASES-99, Boundary-Layer Meteorol., 139(2), 261–281, doi:10.1007/s10546-010-9583-z, 2011.

Wesely, M. L.: Parameterization of surface resistances to gaseous dry deposition in regional-scale numerical models, Atmos. Environ., 23(6), 1293–1304, doi:https://doi.org/10.1016/0004-6981(89)90153-4, 1989.

Xie, B., Fung, J. C. H., Chan, A. and Lau, A.: Evaluation of nonlocal and local planetary boundary layer schemes in the WRF model, J. Geophys. Res. Atmos., 117(12), 1–26, doi:10.1029/2011JD017080, 2012.

Yan, S., Zhu, B., Zhu, T., Shi, C. and Liu, D.: The Effect of Aerosols on Fog Lifetime : Observational Evidence and Model Simulations Geophysical Research Letters, , 1–10, doi:10.1029/2020GL091156, 2021.